# Enhancing LLM Reasoning for Time Series Classification by Tailored Thinking and Fused Decision

## Abstract

The reasoning capabilities of large language models (LLMs) have significantly advanced their performance by enabling in-depth understanding of diverse tasks. With growing interest in applying LLMs to the time series domain, this has proven nontrivial, as evidenced by the limited efficacy of straightforwardly adapting text-domain reasoning techniques. Although recent work has shown promise in several time series tasks, further leveraging advancements in LLM reasoning remains under-explored for time series classification (TSC) tasks, despite their prevalence and significance in many real-world applications. In this paper, we propose `ReasonTSC`, a novel framework designed to effectively leverage LLM reasoning for time series classification through both a multi-turn reasoning and a fused decision-making strategy tailored to TSC. Rather than straightforwardly applying existing reasoning techniques or relying solely on LLMs' built-in reasoning capabilities, `ReasonTSC` first steers the model to think over the essential characteristics of time series data. Next, it integrates predictions and confidence scores from plug-in classifiers, e.g., domain-specific time series models, as in-context examples. Finally, `ReasonTSC` guides the LLM through a structured reasoning process: it evaluates the initial assessment, backtracks to consider alternative hypotheses, and compares their merits before arriving at a final classification. Extensive experiments and systematic ablation studies demonstrate that `ReasonTSC` consistently outperforms both existing time series reasoning baselines and plug-in models, and is even capable of identifying and correcting plug-in models' false predictions. The code for `ReasonTSC` is available at https://anonymous.4open.science/r/ReasonTSC-B737.

## 1 Introduction

Time series classification (TSC) is a fundamental task with wide applications across diverse areas, including healthcare (Wang et al., 2024b; Liu et al., 2024c; An et al., 2023), finance (Liang et al., 2023; Majumdar & Laha, 2020), speech recognition (Yang et al., 2021), and so on (Gupta et al., 2020; Wang et al., 2022). The astounding performance of large language models (LLMs), especially boosted by recent advancements in their reasoning capabilities as epitomized by ChatGPT-o1 (Hurst et al., 2024; Achiam et al., 2023), Deepseek-R1 (Guo et al., 2025), Gemini-2.5-Pro (Team et al., 2023; 2024), has sparked surging demand for leveraging them in domains well beyond the pure natural language processing (NLP) domain. The time series (TS) domain is no exception to such fevered explorations, with existing research promisingly discovering that LLMs have the capability to understand essential TS data characteristics, such as trend, cyclic behavior, stationarity, amplitude, rate of change, and outlier (Chen et al., 2024; Deng et al., 2024). Consequently, a variety of methods have been proposed to exploit LLMs for TS tasks (Prabhakar Kamarthi & Prakash, 2024; Liu et al., 2024g;b; Ekambaram et al., 2024), with a predominant focus on forecasting tasks that align more naturally with the autoregressive generation behavior of LLMs (Woo et al., 2024; Zhou et al., 2023; Jin et al., 2024a; Lu et al., 2024). There are also efforts exploring LLMs for anomaly detection (Zhou & Yu, 2024; Zhou et al., 2023; Dai et al., 2024), imputation (Wang et al., 2025; 2024d; Goswami et al., 2024), and nascent but growing attempts at classification (Wen et al., 2025; Feofanov et al., 2025; Tao et al., 2024).

Propelled by the promise that advanced reasoning techniques can provide enhanced performance through in-depth understanding of complex tasks (Wang et al., 2024a; Wei et al., 2022), it has become a new frontier to leverage the reasoning capabilities of LLMs in the time series domain (Chow et al., 2024; Merrill et al., 2024; Jin et al., 2024b). However, straightforwardly applying existing reasoning techniques, despite their effectiveness in the NLP domain, to the time series domain leads to minimal performance gains, suggesting it is a nontrivial task to leverage LLMs for effective reasoning about TS. For example, REC4TS (Liu et al., 2025b) reports that reasoning LLMs (i.e., having built-in reasoning enhancements acquired during post-training), Chain-of-Thought (CoT), and self-correction all fail to consistently improve forecasting accuracy, with only self-consistency yielding modest gains. (Merrill et al., 2024) assess three reasoning styles, i.e., etiological reasoning, question answering, and context-aided forecasting, and find that the first two offer negligible benefit while the third produces only modest improvements when given highly relevant context in the form of descriptive text. TimeCAP (Lee et al., 2025) introduces a novel framework that enhances time-series event prediction by employing two specialized LLM agents. A multi-modal encoder further refines the process by retrieving relevant in-context examples to optimize the predictor's prompts. Other authors conclude that introducing a visual module for understanding visualized TS patterns is essential for effective reasoning (Kong et al., 2025; Chen et al., 2025). (Chow et al., 2024) and (Xie et al., 2024) harness LLMs' reasoning only after incorporating time series as an additional modality, whereby they train a dedicated encoder to convert TS into embeddings that are then fed to the LLM alongside text token embeddings. In particular, (Liu et al., 2025a) shows that vanilla CoT cannot even outperform random guessing, and that in-context learning can absurdly underperform no-context baselines. They also end up resorting to visualizing TS data to have effective reasoning and obtain performance improvement.

**Research Gap.** At first glance, these evaluations seem to conclude that neither LLMs with inference-time reasoning techniques such as CoT and in-context illustration nor even reasoning LLMs with built-in reasoning enhancements are capable of effective reasoning for time series tasks. This makes the multimodal and specialized encoder training approaches appear indispensable to enable LLMs to substantively understand and reason about TS tasks. However, this tentative conclusion somewhat contradicts existing evidence proving that LLMs can comprehend fundamental TS patterns (Gruver et al., 2023; Tan et al., 2024; Liu et al., 2024a), based on which they should be able to grasp essential TS task characteristics for sophisticated reasoning without relying on auxiliary vision modules or specialized encoders. Even more perplexing is the observation that providing LLMs with in-context examples (Liu et al., 2025a), despite providing additional task-relevant information, often degrades classification accuracy rather than improving it, implying that current in-context strategies are ill-suited to TS reasoning. These contradictory phenomena raise the following tempting research questions (RQ):

**RQ1:** Is it possible to steer the reasoning process of LLMs to elicit their built-in understanding of time series patterns for effective reasoning?

**RQ2:** Is there a strategy suitable for fusing in-context knowledge into the LLMs' reasoning process to enhance prediction performance?

**Our work.** In this paper, we focus on the time series classification task and answer both research questions in the affirmative by proposing `ReasonTSC`, which entails a thinking procedure tailored for time series (RQ1) and a fused decision strategy effectively exploiting in-context examples (RQ2). **Tailored thinking:** We posit that the ineffectiveness of existing LLMs' reasoning may stem from the fact that straightforwardly applying NLP-domain reasoning techniques or relying on the reasoning LLMs' built-in reasoning enhancements is insufficient to guide the model to spontaneously think over TS data characteristics. LLMs acquire reasoning skills through training on mathematics and coding tasks (Liu et al., 2024e), but rarely on time series tasks, which causes them to lack the spontaneous tendency to reason about TS patterns. Motivated by this, we propose a multi-turn thinking procedure tailored to TSC, featuring a more tightly guided reasoning strategy. `ReasonTSC` explicitly asks LLM to identify and think about key TS data patterns. Furthermore, after the LLM provides a preliminary prediction, `ReasonTSC` explicitly prompts it to reconsider whether alternative answers might be more feasible, drawing on a backtracking strategy shown to be useful in the NLP domain. **Fused decision:** When few-shot examples are available for in-context knowledge, we devise a fused decision strategy. First, rather than directly feeding LLMs with context information in the form of text descriptions of the data characteristics, we find it is more effective to present few-shot examples from different classes and prompt the model to autonomously compare their TS data patterns. Moreover,

instead of visualizing TS data for a vision module or training a specialized encoder for TS embeddings, we propose to introduce off-the-shelf and amply available time series foundation models (TSFM) into the reasoning process. This approach offers two key strengths: 1) TSFMs are pretrained on vast time series datasets, enabling them to provide more relevant information than vision module (e.g., ViT) trained on images or TS encoders trained on much smaller TS datasets; 2) TSFMs are generally more lightweight than vision foundation models, e.g., fusing MOMENT (341M parameters) with Chronos (710M parameters) substantially boosts the classification accuracy of LLMs. To integrate TSFM outputs into the LLM's reasoning pipeline, `ReasonTSC` explicitly interprets TSFM's prediction and confidence score, then makes a fused decision by taking both the interpretation of TSFM's outputs and the LLM's own analysis of TS patterns into the reasoning process.

We conduct extensive experiments and systematic ablation studies on 15 TS benchmark datasets, using 2 TSFMs and 16 mainstream LLMs to validate the effectiveness of `ReasonTSC`. Our key findings are: 1) `ReasonTSC` achieves averagely 90% performance improvement compared with a vanilla CoT prompt adopted by existing work (Zhou & Yu, 2024), demonstrating that its tailored reasoning procedure comprehends TS characteristics more thoroughly, thereby solving the classification task more effectively; 2) When applied across 16 mainstream LLMs, `ReasonTSC` consistently outperforms plain CoT prompting, suggesting its broad compatibility; 3) Notably, `ReasonTSC` can sometimes overturn TSFM's incorrect predictions, indicating that its elicited thinking from LLMs regarding TS characteristics involves a nuanced and in-depth analysis essential for accurate predictions. In summary, the main contributions of this paper are:

- We critically investigate the emerging paradigm of leveraging LLMs' reasoning for the time series domain and posit that LLMs are capable of effective reasoning, contrary to prior conclusions that they cannot achieve performance gains through time series reasoning;
- Through the lens of time series classification, we prove it is indeed possible to leverage LLMs for effective time series reasoning by proposing `ReasonTSC`, a novel framework featuring a tailored multi-turn thinking procedure to explicitly steer models to analyze key TS patterns and alternative predictions, alongside a fused decision strategy to enhance in-context example utility;
- We conduct extensive experiments and systematic ablation studies on 15 datasets, with 2 TSFM from different categories, across 16 mainstream LLMs to verify the effectiveness of `ReasonTSC`.

The *Supplementary Material* provides source code and an Appendix with detailed related work, experiment settings, and additional results, and further details of the proposed method.

## 2 THE PROPOSED REASONTSC

### 2.1 PROBLEM FORMULATION

Let $\mathcal{D} = \{(x_i, y_i), i = 0, 1, ..., N - 1\}$ denotes a time series dataset with N samples, where $x_i \in \mathcal{R}^{m \times w}$ is a sample with $m$ variables measured for $w$ steps, $y_i \in \{1, 2, ..., C\}$ is the corresponding label with C be the number of classes. The classical time series classification problem is to train a classification model on the training dataset $\mathcal{D}^{train}$, which can predict the labels of samples in the testing dataset $\mathcal{D}^{test}$,

$$\hat{y}_t = f(x_t), t = 0, 1, ..., M - 1, \tag{1}$$

where M is the number of samples in the testing dataset. In this work, we propose to adopt a reasoning LLM to enhance the time series classification task.

Let $f_M$ be a reasoning language model that consists of a series of rationales obtained on condition of the time series $\mathcal{X}_j$ and tailored prompts $\phi(\mathcal{X}_j)$ in a multi-turn manner, which is applied to enhance various time series classification tasks.

$$r_j \simeq p_\theta(r_j | r_{j-1}, \mathcal{X}_j, \phi(\mathcal{X}_j)), j = o, 1, ..., J - 1; \tag{2}$$

$$f_M \simeq p_\theta(r_0, r_1, ..., r_{J-1}, \mathcal{X}, \phi(\mathcal{X})); \tag{3}$$

$$\hat{y}_t = f_M(x_t, \psi(x_t)), t = 0, 1, ..., M - 1, \tag{4}$$

where $J$ is the number of reasoning turns/steps, $\phi(\mathcal{X}_j)$ is the tailored prompt based on the corresponding input time series samples for the $jth$ reasoning turn/step, $p_\theta$ is an LLM, $f_M$ is the final output based on all the intermediate rationales and input samples, $x_t$ is the testing sample, M is the number of testing samples, and $\psi(x_t)$ is the tailored prompt designed for the testing time series sample $x_t$.

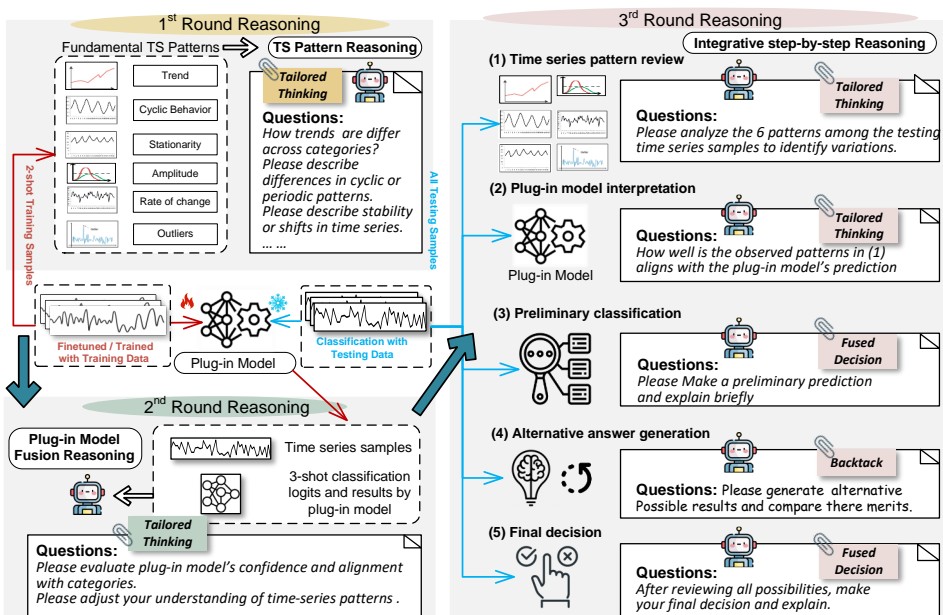

Figure 1: Architecture of the proposed ReasonTSC framework.

## 2.2 THE REASONTSC FRAMEWORK

As illustrated in Figure 1, the proposed ReasonTSC framework comprises three reasoning turns: (1) TS Pattern Reasoning, where the language model is asked to think about the general patterns of time series data; (2) Plug-in Model Fusion Reasoning, where the classification logits of a fine-tuned/pretrained domain-specific time series model is plugged in the reasoning paradigm to enhance LLM's understanding of the TSC task; and (3) Integrative Step-by-step Reasoning, where the reasoning paradigm is conducted step-by-step by evaluating the initial assessment, backtracking alternative hypotheses, and comparing different answers before reaching a final decision.

**TS Pattern Reasoning.** As mentioned in Section 1, LLM can learn to generate realistic time series by analyzing several fundamental time series characteristics such as trend, amplitude, stationarity, and so on (Cai et al., 2024; Potosnak et al., 2024), which indicates that LLM can better understand the intrinsic time series patterns by thinking about these traits.

Trend: A persistent, long-term directional movement (upward/downward) in the time series. It reveals fundamental shifts in data behavior at the macro-level.

**Cyclic behavior**: Repeating patterns or periodic fluctuations. It enables the detection of seasonal or cyclical variations.

**Stationarity**: The stability of time-invariant statistical properties (mean, variance) or their shifts. It is essential for assessing the underlying structure of time series.

**Amplitude**: The maximal deviation magnitude during fluctuations. It quantifies the intensity of variations in the data.

**Rate of change**: The speed at which the data changes (rapid/moderate/slow). It characterizes the temporal dynamics of the time series.

**Outliers**: Data points that deviate significantly from normal values. It may indicate anomalies and data quality issues.

Thus, for the ReasonTSC framework, we first aim to obtain the LLM rationales by answering questions in terms of time series fundamental traits (Xie et al., 2024; Merrill et al., 2024). To be specific, 2-shot time series samples are randomly selected per category from the training set. The LLM is prompted to compare the differences among various categories in terms of the selected fundamental traits. We also include domain-specific knowledge in the prompts and encourage the adopted LLM to decompose a series into semantically meaningful segments to enhance its understanding (Deng et al., 2024). Please refer to the Appendix C for complete prompts.

**Plug-in Model Fusion Reasoning.** According to (Yang et al., 2024b), classification results by a small model could enhance LLM's ability on domain-specific tasks. Here, we propose to plug in a

Table 1: Classification accuracy (%). MOMENT is plugged in for `ReasonTSC`.

| Model | Dist. TW | Mid. TW | Mid. OA | Elec. | Med. Img | BME | Arr. Hd | Dod. LD |
|---|---|---|---|---|---|---|---|---|
| MOMENT (*reference and fused TSFM*) | 62.59 | 51.30 | 60.39 | 57.89 | 76.97 | 74.00 | 65.71 | 31.17 |
| Vanilla CoT (GPT-4o-mini) | 33.81 | 23.38 | 41.56 | 36.84 | 9.87 | 42.34 | 45.14 | 15.58 |
| `ReasonTSC` (GPT-4o-mini) | 63.31 | 53.40 | 61.04 | 58.55 | 77.63 | 77.33 | 68.00 | 31.17 |
| Improvement vs. Vanilla | +87.25% | +128.40% | +46.87% | +58.93% | +686.52% | +82.64% | +50.64% | +100.06% |
| Improvement vs. TSFM | +1.15% | +4.09% | +1.08% | +1.14% | +0.86% | +4.50% | +3.49% | +0.00% |
| Vanilla CoT (Llama-3.3-70B-instruct) | 33.10 | 41.24 | 31.17 | 46.71 | 13.16 | 59.00 | 42.36 | 31.81 |
| `ReasonTSC` (Llama-3.3-70B-instruct) | 63.31 | 53.95 | 61.04 | 61.18 | 77.63 | 84.00 | 66.86 | 36.36 |
| Improvement vs. Vanilla | +91.27% | +30.82% | +95.83% | +30.98% | +489.89% | +42.37% | +57.84% | +14.30% |
| Improvement vs. TSFM | +1.15% | +5.17% | +1.08% | +5.68% | +0.86% | +13.51% | +1.75% | +16.65% |
| Vanilla CoT (DeepSeek-R1) | 52.52 | 47.08 | 33.11 | 51.98 | 37.17 | 76.66 | 54.86 | 28.57 |
| `ReasonTSC` (DeepSeek-R1) | **65.71** | **57.42** | **63.64** | **67.11** | **80.26** | 82.67 | **69.14** | **38.96** |
| Improvement vs. Vanilla | +25.11% | +21.96% | +92.21% | +29.11% | +115.93% | +7.84% | +26.03% | +36.37% |
| Improvement vs. TSFM | +4.98% | +11.93% | +5.38% | +15.93% | +4.27% | +11.72% | +5.22% | +24.99% |

| Model | CBF | Rkt. Spt | ERing | Nt.Ops | Lbr. | Eplp. | Pen. | Avg |
|---|---|---|---|---|---|---|---|---|
| MOMENT (*reference and fused TSFM*) | 66.00 | 59.21 | 72.59 | 65.56 | 48.49 | 88.40 | 85.62 | 64.39 |
| Vanilla CoT (GPT-4o-mini) | 45.67 | 34.26 | 36.67 | 38.61 | 22.78 | 51.45 | 21.92 | 33.33 |
| `ReasonTSC` (GPT-4o-mini) | 65.33 | **67.76** | **74.81** | 65.56 | 48.89 | 89.13 | 86.30 | 65.88 |
| Improvement vs. Vanilla | +43.05% | +97.78% | +104.01% | +69.80% | +114.62% | +73.24% | +293.7% | +135.83% |
| Improvement vs. TSFM | -1.02% | +14.44% | +3.06% | +0.00% | +0.82% | +0.83% | +0.79% | +2.31% |
| Vanilla CoT (Llama-3.3-70B-instruct) | 47.67 | 39.48 | 51.11 | 38.61 | 25.83 | 55.44 | 23.63 | 38.69 |
| `ReasonTSC` (Llama-3.3-70B-instruct) | 73.33 | 61.84 | 74.07 | 66.67 | 51.11 | 89.86 | **86.99** | 67.21 |
| Improvement vs. Vanilla | +62.22% | +56.64% | +44.92% | +72.68% | +97.87% | +62.09% | +268.13% | +101.19% |
| Improvement vs. TSFM | +11.11% | +4.44% | +2.04% | +1.69% | +5.40% | +1.65% | +1.60% | +4.38% |
| Vanilla CoT (DeepSeek-R1) | 65.00 | 47.04 | 55.56 | 46.11 | 38.89 | 63.41 | 40.76 | 49.25 |
| `ReasonTSC` (DeepSeek-R1) | **74.00** | 63.16 | 74.07 | **67.78** | **55.00** | **91.30** | 86.30 | **69.10** |
| Improvement vs. Vanilla | +13.85% | +34.27% | +33.32% | +47.00% | +41.42% | +43.98% | +111.73% | +45.34% |
| Improvement vs. TSFM | +12.12% | +6.67% | +2.04% | +3.39% | +13.43% | +3.28% | +0.79% | +7.31% |

task-specific classifier to obtain further rationales about the TSC tasks by integrating the classification logits. Specifically, a task-specific time series classifier is first trained on the training dataset. Then, 3-shot time series samples are randomly selected from the training set and fed to the trained classifier to obtain its classification logits and decision confidence. The logits, confidence, the ground truth labels, and the basic information (e.g., its training accuracy) of the trained task-specific plug-in model are fused as auxiliary references for the LLM to understand the TSC task. The LLM is asked to analyze cases where the plug-in model correctly or incorrectly identifies different classes to refine its understanding of how to conduct the TSC task. Please refer to the Appendix C for complete prompts.

**Integrative Step-by-step Reasoning.** For the third reasoning turn, we concatenate each testing time series sample with its corresponding predicted label and confidence scores from the plug-in model as input to the reasoning LLM. Rather than simply adopting the generic "think step by step" prompt prefix, we design a tailored CoT approach for the TSC task. The reasoning LLM, with its ability gained in the first two turns, is asked to analyze the patterns of the testing sample and the classification results provided by the plug-in model. Based on this analysis, the reasoning LLM generates a preliminary prediction with a supporting rationale. Then, the LLM is asked to backtrack and explore alternative predictions and systematically compare their merits against the initial assessment. Finally, the reasoning LLM synthesizes all evidence to generate a refined final classification decision. Please refer to the Appendix C for complete prompts.

# 3 EXPERIMENTS

## 3.1 EXPERIMENTAL SETTINGS

**Plug-in domain-specific time series models.** We select two prominent time series foundation models as the plug-in classifiers: (1) MOMENT (Goswami et al., 2024), a T5-based encoder-only model, which is fully fine-tuned with our training data. (2) Chronos (Ansari et al., 2024) is an encoder-decoder model primarily designed for TS forecasting, whose pretrained encoder is adopted to extract time series embeddings for training an SVM-based classifier with the training data.

**Reasoning LLMs.** The main body of experiments is conducted with three primary LLMs—GPT-4o-mini, Llama-3-70B-Instruct, and DeepSeek-R1, covering different parameter scales and reasoning

Table 2: Classification accuracy (%). Chronos is plugged in for `ReasonTSC`.

| Model | Dist. TW | Mid. TW | Mid. OA | Elec. | Med. Img | BME | Arr. Hd | Dod. LD |
|---|---|---|---|---|---|---|---|---|
| Chronos (*reference and fused TSFM*) | 60.43 | 57.79 | 52.60 | 46.71 | 65.39 | 76.00 | 48.57 | 55.84 |
| Vanilla CoT (GPT-4o-mini) | 33.81 | 23.38 | 41.56 | 36.84 | 9.87 | 42.34 | 45.14 | 15.58 |
| `ReasonTSC` (GPT-4o-mini) | 61.15 | 57.79 | **57.14** | 47.39 | 69.74 | 78.00 | **54.29** | 58.44 |
| Improvement vs. Vanilla | +80.86% | +147.18% | +37.49% | +28.64% | +606.59% | +84.22% | +20.27% | +275.10% |
| Improvement vs. TSFM | +1.19% | +0.00% | +8.63% | +1.46% | +6.65% | +2.63% | +11.78% | +4.66% |
| Vanilla CoT (Llama-3.3-70B-instruct) | 33.10 | 41.24 | 31.17 | 46.71 | 13.16 | 59.00 | 42.36 | 31.81 |
| `ReasonTSC` (Llama-3.3-70B-instruct) | 64.03 | 59.09 | 53.90 | 48.03 | 71.05 | **86.00** | 50.29 | 57.14 |
| Improvement vs. Vanilla | +93.44% | +43.28% | +72.92% | +2.83% | +439.89% | +45.76% | +18.72% | +79.63% |
| Improvement vs. TSFM | +5.96% | +2.25% | +2.47% | +2.83% | +8.66% | +13.16% | +3.54% | +2.33% |
| Vanilla CoT (DeepSeek-R1) | 52.52 | 47.08 | 33.11 | 51.98 | 37.17 | 76.66 | 54.86 | 28.57 |
| `ReasonTSC` (DeepSeek-R1) | **64.75** | **61.69** | 54.55 | **53.95** | **73.03** | 85.33 | **54.29** | **62.34** |
| Improvement vs. Vanilla | +23.29% | +31.03% | +64.75% | +3.79% | +96.48% | +11.31% | -1.04% | +118.20% |
| Improvement vs. TSFM | +7.15% | +6.75% | +3.71% | +15.50% | +11.68% | +12.28% | +11.78% | +11.64% |

| Model | CBF | Rkt. Spt | ERing | Nt.Ops | Lbr. | Eplp. | Pen. | Avg |
|---|---|---|---|---|---|---|---|---|
| Chronos (*reference and fused TSFM*) | 90.89 | 54.61 | 53.33 | 62.22 | 42.22 | 91.30 | 68.49 | 61.76 |
| Vanilla CoT (GPT-4o-mini) | 45.67 | 34.26 | 36.67 | 38.61 | 22.78 | 51.45 | 21.92 | 33.33 |
| `ReasonTSC` (GPT-4o-mini) | 89.33 | 55.26 | 54.81 | 63.89 | 41.67 | 91.30 | 65.75 | 63.06 |
| Improvement vs. Vanilla | +95.60% | +61.29% | +49.47% | +65.48% | +82.92% | +77.45% | +199.95% | +127.50% |
| Improvement vs. TSFM | -1.72% | +1.19% | +2.78% | +2.68% | -1.30% | +0.00% | -4.00% | +2.10% |
| Vanilla CoT (Llama-3.3-70B-instruct) | 47.67 | 39.48 | 51.11 | 38.61 | 25.83 | 55.44 | 23.63 | 38.69 |
| `ReasonTSC` (Llama-3.3-70B-instruct) | **95.33** | 55.26 | 57.04 | 66.67 | 45.00 | 92.03 | **69.18** | 64.67 |
| Improvement vs. Vanilla | +99.98% | +39.97% | +11.60% | +72.68% | +74.22% | +66.00% | +192.76% | +90.25% |
| Improvement vs. TSFM | +4.89% | +1.19% | +6.96% | +7.15% | +6.58% | +0.80% | +1.01% | +4.71% |
| Vanilla CoT (DeepSeek-R1) | 65.00 | 47.04 | 55.56 | 46.11 | 38.89 | 63.41 | 40.76 | 49.25 |
| `ReasonTSC` (DeepSeek-R1) | 93.33 | **64.28** | **62.96** | **67.78** | **57.22** | **94.93** | 61.64 | **67.47** |
| Improvement vs. Vanilla | +43.58% | +36.65% | +13.32% | +47.00% | +47.13% | +49.74% | +51.23% | +42.43% |
| Improvement vs. TSFM | +2.68% | +17.71% | +18.06% | +8.94% | +35.53% | +3.98% | -10.00% | +9.57% |

Table 3: Results of `ReasonTSC`'s classification overrides against plug-in models. The Overriden (%) shows the percentage of classification results that are different from those by plug-in models. The Override Accuracy (%) shows the rate of correct classification results among these overrides.

| | Overriden (%) | | | Override Accuracy (%) | | |
|---|---|---|---|---|---|---|
| | MOMENT | Chronos | Average | MOMENT | Chronos | Average |
| `ReasonTSC` (GPT-4o-mini) | 2.77 | 5.68 | 4.23 | 65.34 | 29.37 | 47.36 |
| `ReasonTSC` (Llama-3.3-70b-instruct) | 4.23 | 6.00 | 5.12 | 83.30 | 71.51 | 77.41 |
| `ReasonTSC` (Deepseek-R1) | 9.42 | 14.36 | 11.89 | 68.47 | 62.88 | 65.68 |

training techniques. To further investigate how reasoning LLMs can enhance TSC tasks, we also evaluate the performance of `ReasonTSC` with six other mainstream LLMs on three selected UCR/UEA datasets, including ChatGPT, Claude, Gemini, Qwen (Bai et al., 2023; Yang et al., 2024a), Llama (Grattafiori et al., 2024), and Grok, with a fixed temperature parameter of 0.2.

**Datasets.** We select 15 datasets from the UCR/UEA classification archive (Dau et al., 2019; Bagnall et al., 2018) that are commonly used for benchmarking classification algorithms. The selected datasets cover a class range from 3 to 15, and include 80 to 288 samples per category. Under the 2-shot setting, the total input length varies between 160 and 4032, corresponding to approximately 1.8k to 12k tokens. We use the first dimension of the multivariate UEA datasets to address the token limit restrictions imposed by LLM input queries. Given the typically long sequence lengths of time series samples, we retain values to three decimal places to optimize context window usage. Please refer to Appendix D for details about LLMs and datasets.

## 3.2 MAIN RESULTS

As shown in Tables 1 and 2, the vanilla CoT, which is merely prompted with *please think step by step* before the question, yields consistently low accuracy values across different LLMs (Zhou & Yu, 2024). This observation reveals that LLMs cannot enhance TSC tasks by adopting their built-in reasoning capabilities with CoT. On the contrary, `ReasonTSC` achieves substantial performance improvements (+20%∼ +600%, average 90%) by incorporating a tailored thinking and fused decision strategy. Notably, on datasets with over 11k tokens (Arr.Hd, Eplp, and Dod.LD), `ReasonTSC` with DeepSeek achieves performance improvements of 26.03%, 43.98%, and 36.37%, respectively, compared to vanilla CoT. With more scrutiny to compare `ReasonTSC` and the plug-in models, `ReasonTSC` out-

performs the plug-in models across almost all the tested datasets. Specifically, `ReasonTSC` with DeepSeek as the reasoning language model surpasses the plug-in model MOMENT by over 10% on six datasets, including substantial performance improvement by 24.99% on DodgerLoopDay (Dod.LD) and 15.93% on ElectricDevices (Elec.). It is worth mentioning that the plug-in models are fine-tuned/trained on the whole training dataset, while the `ReasonTSC` is only shown with two samples per category, indicating the efficiency of the proposed reasoning strategy. Further analysis of `ReasonTSC`'s stability, comparisons with traditional full-shot baselines, and generalization analysis are provided in Appendix B.

To further investigate the proposed `ReasonTSC`'s reasoning capabilities, we show the average override rates of `ReasonTSC` compared with plug-in models as shown in Table 3. `ReasonTSC` with DeepSeek exhibits an override rate of 11.89% on average, which is higher than that by ReasonTS (Llama) (5.12%) and `ReasonTSC` (GPT) (4.23%). Regarding override accuracy, `ReasonTSC` (Llama) and `ReasonTSC` (DeepSeek) achieve average override accuracy of 77.41% and 65.68%, respectively. This suggests that `ReasonTSC` can effectively leverage LLMs' understanding of time series patterns through multi-turn reasoning to correct incorrect predictions by plug-in models.

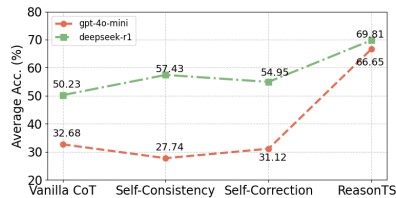

Figure 2: Average performance of ReasonTSC with other reasoning techniques on six UCR/UEA datasets. The number of inference rounds of self-consistency and self-correction is set to 3.

We further compare the performance of `ReasonTSC` with other mainstream reasoning techniques (Liu et al., 2025b). Self-Consistency (Wang et al., 2023) generates multiple reasoning paths in parallel and selects the most consistent result, while Self-Correction (Madaan et al., 2023) iteratively refines the model's output through self-feedback. Figure 2 indicates that naively applying these reasoning techniques to leverage LLM's inherent reasoning ability does not achieve satisfactory performance. In contrast, `ReasonTSC` enhances time-series understanding by integrating LLM's reasoning ability with TSFM's knowledge, demonstrating clear advantages over existing baseline approaches.

Additionally, we also evaluate the proposed `ReasonTSC` with other mainstream LLMs as its reasoning language models on three datasets. As illustrated in Figure 3, the horizontal black dashed line marks the performance of the plug-in model MOMENT. In Figure 3 (a), we compare `ReasonTSC`'s performance in terms of the model sizes of different language models. Here, `ReasonTSC`'s performance does not show an obvious correlation with the sizes and architectures of language models. On

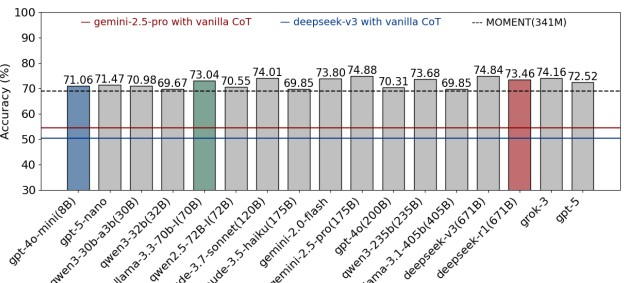

Figure 3: Average performance of `ReasonTSC` with mainstream LLMs as reasoning language models on three selected UCR/UEA datasets (Mid.OA, BME, and ERing).

the other hand, Gemini-2.5-pro (175B parameters) and Deepseek-v3 (671B parameters) achieve the best and second-best performance. The red and blue solid lines represent the performance of Vanilla CoT reasoning with Gemini-2.5-pro and Deepseek-v3, respectively. It is shown that even for the recently newly released LLMs with strong reported built-in reasoning ability, the proposed `ReasonTSC` shows much performance improvement over the Vanilla CoT reasoning strategy. Please refer to Appendix B.7 for complete experimental results.

### 3.3 ANALYSIS OF KEY THINKING STEPS

**Thinking TS patterns.** In the first round of reasoning, `ReasonTSC` thinks about the fundamental TS patterns by showing few-shot training samples of each category. We examine how the number of few-shot examples affects reasoning performance. As shown in Figure 4, the performance of `ReasonTSC` with GPT-4o-mini is relatively stable from 1-shot to 5-shot configurations across three datasets, with only slight degradations. Notably, the 2-shot configuration already yields satisfactory results, demonstrating `ReasonTSC`'s robustness in steering LLMs for TSC tasks.

**Backtracking.** During the integrative step-by-step reasoning process (third reasoning turn), the *alternative answer generation* step guides `ReasonTSC` to backtrack to consider alternative hypotheses and compares their merits before arriving at a final classification decision. Figure 5 illustrates the counts of cases where `ReasonTSC` ultimately adopts alternative candidates in their final predictions. `ReasonTSC` with Llama shows higher sensitivity than `ReasonTSC` s with GPT and DeepSeek, where 58 successful corrections out of 109 alternative adoptions are presented. `ReasonTSC` s with DeepSeek and GPT present successful correction rates of 75% and 42.31%, respectively. This reveals that with the step-by-step integrative reasoning strategy, `ReasonTSC` could comprehensively consider TS patterns and plug-in model's auxiliary information, and correct its preliminary decision.

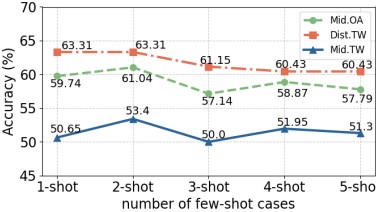
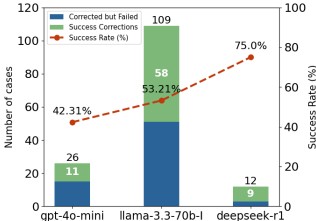

Figure 4: Performance of `ReasonTSC` with GPT in terms of the number of few-shot examples.

Figure 5: Effectiveness of the *alternative answer generation* step in the 3rd turn of reasoning.

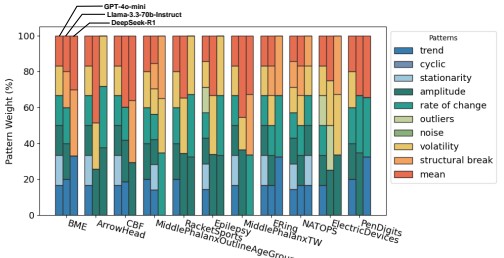
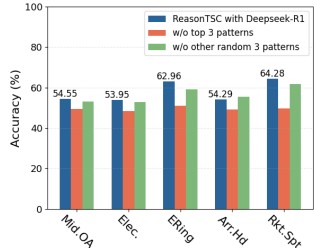

Figure 6: Evaluation of `ReasonTSC`'s ability to reason about time series patterns using real-world datasets. For each of the tested 11 datasets, the predominant patterns identified by GPT-4o-mini, Llama3.3-70b-instruct, and DeepSeek-R1 are shown in the bars in a left-to-right order.

Figure 7: Performance comparison of `ReasonTSC` with Deepseek-R1: removing either the top three patterns or three random patterns during the time series pattern reasoning round.

### 3.4 RESEARCH QUESTIONS

#### 3.4.1 TS PATTERN INTERPRETATION (RQ1)

To further answer **RQ1**, we evaluate `ReasonTSC`'s ability to think about time-series patterns in this section. We first construct four synthetic time series datasets, where the first three individually exhibit distinct trend, frequency, and amplitude patterns, while the last one integrates these three patterns. We present each time series sample alongside randomly generated noise sequences in a multiple-choice format, questioning the `ReasonTSC` to identify the sequence with the most discernible patterns. Choice positions are randomized to eliminate positional bias. Notably, `ReasonTSC` s with GPT, Llama, and Deepseek achieve satisfactory accuracy across all the tested datasets, **demonstrating `ReasonTSC`'s ability to generate rationales about fundamental time series patterns**. Details of dataset construction, question design, and related prompts are provided in Appendix E. We further evaluate `ReasonTSC`'s ability to reason about time-series patterns using 11 datasets from the realistic UCR/UEA archives. Here, we prompt the `ReasonTSC` to identify fundamental patterns (*trend, cyclic, stationarity, amplitude, rate of change, outliers, noise, volatility, structural break, and mean shift* (Cai et al., 2024)) mentioned in Section 2. As shown in Figure 6, `ReasonTSC` with GPT-4o-mini consistently identifies similar TS patterns (e.g., trend, amplitude, rate of change, volatility, and mean shift) across all datasets, suggesting it tends to present more generalized interpretations (cannot discern different datasets), which aligns with the final classification performance where it shows relatively lower classification accuracy. On the contrary, `ReasonTSC` with DeepSeek-R1 (which also shows the best overall classification performance) shows superior performance in identifying category-discriminative patterns: it recognizes trend, structural break, and mean shift as distinctive features in the BME dataset, while recognizing amplitude, rate of change, and volatility as predominant in the ArrowHead dataset. Additionally, we conduct an ablation study by either

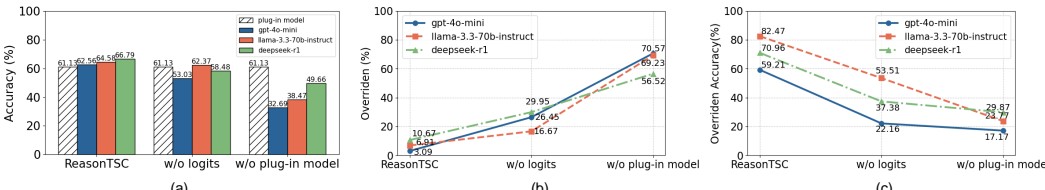

Figure 8: Ablation study of `ReasonTSC` under three configurations: without logits and the whole plug-in model. Three merits are compared under these conditions: classification performance (a), overridden rate (b), and override accuracy (c). All results represent the average performance obtained on 9 selected UCR/UEA subsets.

removing the top three predominant patterns identified by DeepSeek-R1 or three other randomly selected patterns from each subset. As illustrated in Figure 7, removing the top patterns causes a noticeable performance drop in `ReasonTSC` with DeepSeek-R1, whereas removing random patterns yields results comparable to the original `ReasonTSC`. **These observations indicate that a better understanding of the time series patterns could enhance the reasoning process of LLMs and the TSC accordingly**. Details of prompts and corresponding answers are provided in Appendix E.

### 3.4.2 ABLATION OF FUSION STRATEGY (RQ2)

To answer **RQ2**, we conduct ablation studies to evaluate the impact of the fused decision strategy: (1) reasoning about the category-wise confidence scores (logits) of the plug-in model (w/o logits), and (2) the complete outputs (logits & final predictions) of the plug-in model (w/o plug-in model). As illustrated in Figure 8 (a), removing the plug-in model's logits leads to an 8.31% performance decline in `ReasonTSC` with DeepSeek; Completely removing outputs of the plug-in model leads to a significant performance decrease. **This indicates the importance of the fused decision strategy**.

As shown in Figure 8 (b) and (c), the override rates of `ReasonTSC`'s increase while their overall override accuracy decreases with reduced reasoning supports. When the plug-in model's logits are removed, we observe higher override rates and bigger accuracy degradation, which also **shows that the fused decision strategy with the plug-in model enhances `ReasonTSC`'s performance in TSC**. For the w/o plug-in setting, the reported override metrics are computed post hoc by comparing the LLM's predictions with the plug-in's outputs, reflecting the their disagreement and highlighting the effect of the fusion strategy. Please refer to Appendix B for more ablation studies.

### 3.4.3 DECISION INTERPRETATION (RQ1&2)

Since `ReasonTSC` is asked to explain its final decision, we can count for each override case which information drives the model to make different classification results. As shown in Figure 9, `ReasonTSC` with GPT relies on the plug-in model's logits and TS patterns in all the override cases. `ReasonTSC` with Llama and DeepSeek partially rely on the plug-in model's accuracy for their override decisions. `ReasonTSC` with GPT relies on the TS patterns only for the majority of override cases (63.49%). As discussed in Section 3.4.1,

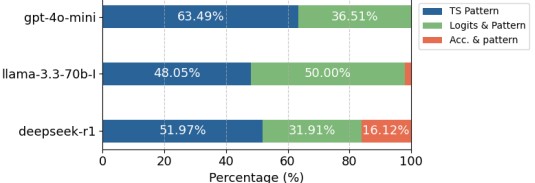

Figure 9: Reasons for `ReasonTSC` override: (i) primary reliance on typical time series patterns, (ii) consideration of both the plug-in model's logits and time series patterns, (iii) combined assessment of the plug-in model's accuracy and time series patterns.

`ReasonTSC` with GPT cannot discern the TS patterns among different categories. Its heavy reliance on the TS patterns for final decision can also explain its relatively low classification performance compared to the other two scenarios (`ReasonTSC` with Llama & DeepSeek). This interpretation analysis shows both TS patterns and plug-in models influence `ReasonTSC`' performance.

## 4 CONCLUSION

This paper has proposed `ReasonTSC`, a novel framework that effectively leverages reasoning LLMs for time series classification through a multi-turn reasoning and fused decision-making strategy. It first guides the LLM to analyze the intrinsic patterns of time series data. It then incorporates predictions

and category-wise confidence scores from the plug-in model as in-context examples to enhance its understanding of the TSC task. Finally, `ReasonTSC` orchestrates a structured reasoning pipeline: the LLM evaluates its initial assessment, backtracks to consider alternative hypotheses, and compares their merits before determining the final classification. Extensive experiments and ablation studies demonstrate that `ReasonTSC` consistently outperforms both LLMs with Vanilla CoT reasoning and plug-in models, and is even capable of identifying plug-in models' false predictions and correcting them accordingly. This reveals significant potential for leveraging reasoning LLMs to enhance time series classification tasks in various domains. However, ReasonTSC remains constrained by the inherent context length limitations of LLMs when processing long time series sequences. Future work could explore alternative tokenization methods to improve time series representation for LLMs.

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

APPENDIX: ENHANCING LLM REASONING FOR TIME SERIES CLASSIFICATION BY TAILORED THINKING AND FUSED DECISION

This appendix contains: 1) Section A: deferred related work; 2) Section B: additional experimental results and analysis; 3) Section C: reasoning details of our proposed `ReasonTSC` framework; 4) Section D: implementation details of `ReasonTSC`, including datasets, adopted LLMs, and adopted time series foundation models; 5) Section E: comprehensive interpretation study of `ReasonTSC` on time series patterns; 6) Section F: limitations and discussions of `ReasonTSC`; 7) Section G: statement on the use of LLMs. Our source code for replicating the experiments can be found in https://anonymous.4open.science/r/ReasonTSC-B737.

## A    RELATED WORK

In addition to the existing works discussed in Section 1, we provide a brief review of related works on leveraging LLM for time series analysis, as follows.

**Time-Series Tasks Using LLMs.** The history of time series analysis can be traced back to signal processing and solving the first-order differential equations (Cryer, 2008). With the advent of LLMs, a variety of works have shown the potential for leveraging LLMs for TS tasks. PromptCast (Xue & Salim, 2023) transforms numerical time series into prompts in a natural language generation manner. (Gruver et al., 2023) demonstrates that LLMs like GPT-3 and LLaMA-2 can perform zero-shot time series extrapolation. (Tang et al., 2025) finds that LLMs perform well in forecasting time series with clear patterns but face challenges with datasets lacking periodicity. (Jin et al., 2024a) proposes to reprogram time series data for general forecasting. GPT4TS (Zhou et al., 2023) fine-tunes the GPT-2 backbone to perform time series analysis tasks such as classification and anomaly detection. (Zhou & Yu, 2024) evaluate LLMs' capabilities for conducting time series anomaly detection and conclude that LLMs can understand trivial time series anomalies. LLMs are adopted in (Wang et al., 2025) to impute the highly sparse remote sensing data, where LLMs are applied to capture the spatiotemporal dependencies buried in data sequences. (Tao et al., 2024) adopts a multi-modal approach to leveraging LLMs for enhancing the time series classification task, which trains two additional encoders to convert time series data into embeddings. They do not explore reasoning techniques at the LLM end. In contrast, the primary goal of our work is to investigate whether dedicated reasoning strategies can enhance LLMs' ability to understand time series tasks without relying on additional visual modalities or the training of specialized encoders. Besides, various Time Series Foundation Models (TSFMs), inspired by the corresponding architectures and pretraining strategies from LLM literature, are pretrained with a large scale of time series data (Goswami et al., 2024; Liu et al., 2024h; Woo et al., 2024; Ansari et al., 2024; Das et al., 2024) targeting to solve common time series analysis tasks, including forecasting, anomaly detection, imputation, and classification.

**Time-Series Tasks Using LLMs Reasoning.** Recently, it has become a new frontier to leverage the reasoning capabilities of LLMs in the time series domain (Chow et al., 2024; Merrill et al., 2024; Jin et al., 2024b). LSTPrompt (Liu et al., 2024a) introduces a Chain-of-Thought (CoT) approach that guides LLMs to decompose forecasting tasks into short-term and long-term subtasks. REC4TS (Liu et al., 2025b) evaluates several reasoning strategies, such as Chain-of-Thought and self-correction, to enhance LLMs in forecasting tasks. However, forecasting metrics like MSE only require LLMs to generate approximate extrapolations, whereas reasoning typically demands that LLMs pinpoint definitive answers through grasping the underlying patterns. Thus, (Zhou & Yu, 2024) investigates the time series anomaly detection capabilities of LLMs, while (An et al., 2024) proposes IoT-LLM to explore the potential for IoT task reasoning. InstructTime (Cheng et al., 2025) redefines time series classification by framing it as a multimodal generative task. TS-Reasoner (Ye et al., 2024) enables precise multi-step time series analysis by integrating LLM-based task decomposition using in-context learning and program-aided execution with self-correcting feedback loops. (Potosnak et al., 2025) and (Potosnak et al., 2024) demonstrate that patch-based Transformers exhibit robust generalization to systematic out-of-distribution scenarios, suggesting their intrinsic reasoning abilities surpass mere pattern memorization. However, (Merrill et al., 2024) evaluates LLMs on tasks such as etiological reasoning and context-aided forecasting, revealing that time-series reasoning remains a critical yet severely underdeveloped direction.

Table 4: Performance comparison between `ReasonTSC` and traditional baselines. Note that all baseline methods are task-specific models trained on the entire training set for **100** epochs, whereas `ReasonTSC` utilizes only a few demonstration examples for in-context learning.

| Model | Dist. TW | Mid. TW | Mid. OA | Elec. | Med. Img | BME | Arr. Hd | Dod. LD |
|---|---|---|---|---|---|---|---|---|
| Full-shot (# Training samples): | 400 | 399 | 400 | 8926 | 381 | 30 | 36 | 67 |
| TimesNet (Wu et al., 2023) | 69.78 | 59.74 | 65.58 | 65.3 | 70.26 | 94.00 | **82.28** | 58.75 |
| Autoformer (Wu et al., 2021) | 64.74 | 57.79 | 64.28 | 66.78 | 62.89 | 84.00 | 55.43 | 32.50 |
| FEDformer (Zhou et al., 2022) | 69.06 | 58.44 | 66.23 | **68.27** | 67.10 | 94.67 | 75.42 | 41.25 |
| iTransformer (Liu et al., 2024f) | 70.50 | **62.98** | 64.28 | 60.03 | 73.16 | **95.33** | 80.00 | 55.00 |
| PatchTST (Nie et al., 2023) | 71.94 | 61.03 | 66.23 | 65.30 | 72.76 | 94.67 | 66.85 | 52.50 |
| LightTS (Campos et al., 2023) | **74.10** | 59.74 | 62.33 | 62.40 | 69.87 | 86.00 | 65.71 | 61.25 |
| DLinear (Zeng et al., 2023) | 69.78 | 61.03 | 64.28 | 47.72 | 58.28 | 84.00 | 69.71 | 55.00 |
| Two-shot (# Training samples): | 12 | 12 | 6 | 14 | 20 | 6 | 6 | 14 |
| `ReasonTSC` (Llama-3.3-70B-instruct) | 64.03 | 59.09 | 61.04 | 61.18 | 77.63 | 86.00 | 66.86 | 57.14 |
| `ReasonTSC` (DeepSeek-R1) | 65.71 | 61.69 | 63.64 | 67.11 | 80.26 | 85.33 | 69.14 | **62.34** |

| Model | CBF | Rkt. Spt | ERing | Nt.Ops | Lbr. | Eplp. | Pen. | Avg |
|---|---|---|---|---|---|---|---|---|
| Full-shot (# Training samples): | 30 | 151 | 30 | 180 | 180 | 137 | 7494 | |
| TimesNet (Wu et al., 2023) | 94.44 | **76.97** | 71.85 | **69.44** | 58.33 | 79.71 | **90.10** | 73.76 |
| Autoformer (Wu et al., 2021) | 42.22 | 73.02 | 51.48 | 51.11 | 48.33 | 65.21 | 88.47 | 60.55 |
| FEDformer (Zhou et al., 2022) | 51.44 | 72.36 | 64.07 | 65.56 | **58.33** | 71.73 | 88.82 | 67.51 |
| iTransformer (Liu et al., 2024f) | 90.55 | 72.36 | 71.48 | 66.67 | 56.11 | 73.18 | 86.70 | 71.88 |
| PatchTST (Nie et al., 2023) | 90.00 | 71.05 | **75.18** | 58.33 | 53.33 | 90.57 | 87.79 | 71.83 |
| LightTS (Campos et al., 2023) | 83.55 | 68.42 | 70.00 | 68.89 | 52.22 | 62.31 | 89.05 | 69.05 |
| DLinear (Zeng et al., 2023) | 78.44 | 57.89 | 68.89 | 65.55 | 35.00 | 36.22 | 71.09 | 61.52 |
| Two-shot (# Training samples): | 6 | 8 | 12 | 12 | 30 | 8 | 20 | |
| `ReasonTSC` (Llama-3.3-70B-instruct) | **95.33** | 61.84 | 74.07 | 66.67 | 51.11 | 92.03 | 86.99 | 70.73 |
| `ReasonTSC` (DeepSeek-R1) | 93.33 | 63.16 | 74.07 | 67.78 | 57.22 | **94.93** | 86.30 | 72.80 |

# B  ADDITIONAL EXPERIMENTAL RESULTS

In this section, we present extended experimental analyses to complement the discussions in Subsections 3.3 and 3.4. Subsection B.1 compares `ReasonTSC`'s few-shot performance with traditional full-shot baselines. Subsection B.2 investigates how the plug-in model's ICL examples affect `ReasonTSC`. Subsection B.3 evaluates the combined effect of `ReasonTSC` compared to individual reasoning stages. Subsection B.4 evaluates the impact of `ReasonTSC`'s tailored Chain-of-Thought on performance improvement. Subsection B.5 analyzes the stability performance of `ReasonTSC` in terms of dataset characteristics, prompt sensitivity, and statistical significance. Subsection B.6 analyzes the generalizability of `ReasonTSC` on anomaly detection and vision-based reasoning tasks. Subsection B.7 provides the full results of `ReasonTSC` with mainstream LLMs on three selected datasets. Subsection B.8 analyzes the inference time and API costs of `ReasonTSC`.

## B.1  COMPARING REASONTSC WITH TRADITIONAL FULL-SHOT BASELINES

We compare the performance of `ReasonTSC` with the retraining/finetuning-from-scratch baselines that train/finetune time series classification models on the entire dataset. Unlike traditional task-specific methods that rely on the entire downstream task's dataset for training and lack generalization, `ReasonTSC` requires only a few in-context learning demonstrations, enabling it to handle diverse time series tasks more effectively. As shown in Table 4, `ReasonTSC` with Llama and DeepSeek achieves comparable performance to traditional baselines in most subsets. Our goal is therefore not to claim direct percentage-to-percentage superiority, but to illustrate that `ReasonTSC`, under a few-shot inference setting, can achieve results competitive with fully trained baselines in most cases. Moreover, since `ReasonTSC` can reason about the plug-in model's prediction behavior, it can outperform the plug-in model when a baseline model is used as the plug-in and correct the false predictions.

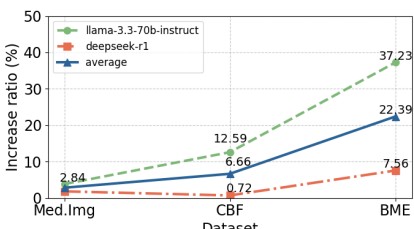 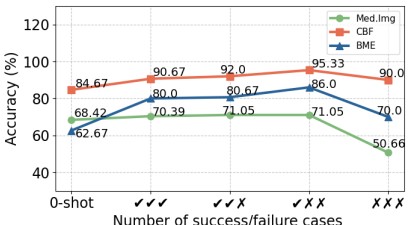

Figure 10: Performance improvement ratio of ICL examples in `ReasonTSC`'s second reasoning round.

Figure 11: Impact of success-failure ratio in ICL examples on `ReasonTSC`'s second reasoning round

## B.2 THE INFLUENCE OF DIFFERENT ICL EXAMPLES IN THE SECOND REASONING ROUND

In the second reasoning round of `ReasonTSC`, the plug-in model's predictions and logits are served as in-context learning (ICL) examples for the LLM. Figure 10 illustrates the performance improvement ratio of `ReasonTSC` when incorporating ICL examples in the second reasoning turn compared to that when ICL examples are omitted. `ReasonTSC` randomly sampled one successful case and two failed cases. Figure 11 further investigates the influence of the success-failure ratio of the selected cases on `ReasonTSC` with Llama. The success case means the plug-in classifier's prediction aligns with the ground truth, while the failed case means otherwise. We test four configurations: all-successful, majority-successful, majority-failed, and all-failed cases.

As illustrated in Figure 10, the prediction behaviors of plug-in models can significantly enhance `ReasonTSC`'s performance on certain datasets. `ReasonTSC` with Llama exhibits increase ratios of 37.23% (BME) and 12.59% (CBF), suggesting that `ReasonTSC` with Llama is more strongly influenced by plug-in model predictions than `ReasonTSC` with DeepSeek.

The success-failure ratio of ICL examples also influences `ReasonTSC`'s performance. By using three success cases as ICL examples, `ReasonTSC` with Llama outperforms the zero-shot setting. Notably, gradually introducing failure cases leads to consistent slight performance improvements, suggesting that the LLM enhances its understanding of time series patterns by analyzing the plug-in model's biased behaviors. Conversely, relying solely on failure cases causes a substantial performance drop. This decline likely arises because negative examples mislead the LLM to reject the plug-in model's valid predictions, yielding counterproductive results.

## B.3 ANALYSIS OF THE COMBINED EFFECT OF REASONTSC'S REASONING STAGES

We evaluate the contribution of the reasoning stage in `ReasonTSC`'s structured multi-round framework. The second and third stages are specifically designed for plug-in model behavior analysis and integrative step-by-step reasoning, respectively. As shown in Figure 12 and Figure 13, removing the second or third round leads to a performance drop in `ReasonTSC` with GPT-4o-mini. This indicates

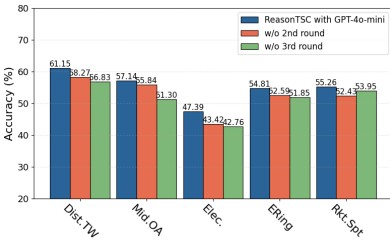 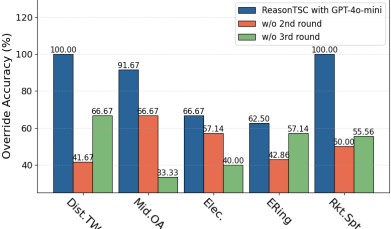

Figure 12: Ablation study of the accuracy of `ReasonTSC` with removal of the 2nd or 3rd reasoning round on five UCR/UEA subsets.

Figure 13: Ablation study of the override accuracy of `ReasonTSC` with removal of the 2nd or 3rd reasoning round on five UCR/UEA subsets.

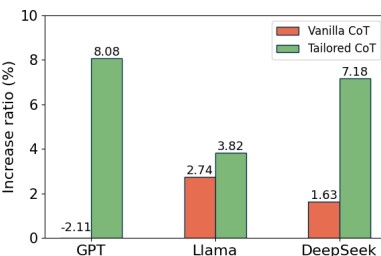 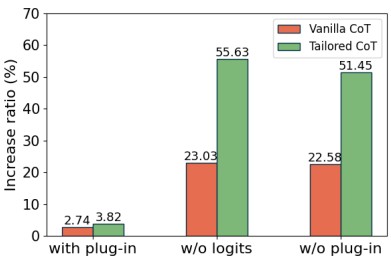

Figure 14: Average performance comparison of `ReasonTSC`'s tailored CoT vs. Vanilla CoT across four UCR/UEA datasets. (Dist.TW, Mid.OA, Med.Img, Dod.LD)

Figure 15: Average performance comparison between `ReasonTSC`'s tailored CoT with Llama and vanilla CoT in terms of with/ without plug-in model across four UCR/UEA datasets.

that the combined effect of the three reasoning stages can surpass the performance of single stage. Furthermore, the significant drop in override accuracy validates that `ReasonTSC` enables the LLM to comprehensively consider time series patterns and auxiliary plug-in model information to correct TSFM's decisions.

### B.4 COMPARING REASONTSC'S TAILORED COT WITH VANILLA COT

The proposed `ReasonTSC` employs a multi-turn reasoning with a fused decision-making strategy tailored to TSC. It steers the LLM to analyze time series patterns in the first reasoning round and guides the LLM to examine the prediction behavior of the plug-in model in the second reasoning round. In the third reasoning round, it reevaluates the initial assessment, backtracks to consider alternative hypotheses, and compares their merits before arriving at a final classification. We conduct an ablation study by removing the tailored CoT in the third reasoning round and replacing it with instruction *please think step by step*. As illustrated in Figure 14, `ReasonTSC` substantially outperforms vanilla CoT, with performance gains of 8.08% for GPT, 3.82% for Llama, and 7.18% for DeepSeek. Note that the performance gains mean the TSC performance improvements by `ReasonTSC` and Vanilla CoT strategies compared to plain LLMs.

We further analyze the performance of `ReasonTSC` with Llama compared to Vanilla CoT by evaluating the impact of removing components from the plug-in model, as depicted in Figure 15. For both methods, removing either the plug-in model's logits or the full model (including predictions and logits) results in significant performance improvements. Notably, our TSC-tailored CoT achieves improvement ratios over twice those of vanilla CoT, reaching 55.63% without logits and 51.45% without the plug-in model. This substantial performance gap further validates the effectiveness of `ReasonTSC`'s customized reasoning strategy for time series classification tasks.

### B.5 PERFORMANCE STABILITY ANALYSIS OF REASONTSC

**Dataset characteristics.** We investigate the impact of three key factors on `ReasonTSC`'s reasoning performance: category count, time series length, and token count, as shown in Figure 16. Regarding the number of classification categories, both `ReasonTSC` with Llama and DeepSeek present stable performance. `ReasonTSC` with GPT exhibits a performance decline as the category count increases, suggesting that a smaller-scale language model faces limitations as the volume of the processed information increases. On the other hand, for sequence lengths less than 80 timestamps, `ReasonTSC` s with Llama and DeepSeek achieve only 3.38% and 8.19% performance improvements, respectively. This is because shorter time series samples contain fewer discernible patterns, which provides less information for the LLM to understand TS. In terms of the number of tokens, `ReasonTSC` with GPT performs best with fewer than 6,000 tokens. `ReasonTSC`with Llama and DeepSeek achieve the best performance with an input token count between 6,000 and 10,000, with improvement ratios of 6.39% and 12.42%, respectively.

**Prompt sensitivity.** To analyze prompt sensitivity, we first task DeepSeek-R1 with paraphrasing the original prompt of `ReasonTSC` while preserving its core meaning. As shown in Table 5, the

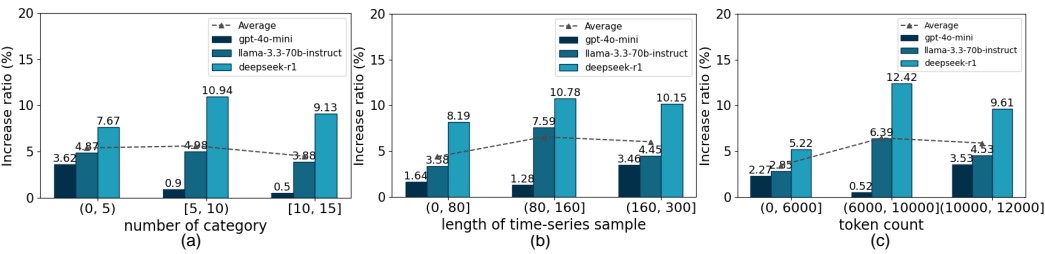

Figure 16: Average performance improvement of `ReasonTSC` compared to TSFMs across all the tested datasets. Three influence factors are considered: category count (a), time series length (b), and token count(c).

performance of `ReasonTSC` with GPT-4o-mini remains consistent under both the original and paraphrased prompts, further demonstrating the robustness of `ReasonTSC` to variations in prompt phrasing.

Table 5: Prompt sensitivity analysis results.

| Dataset | Original (%) | Paraphrased (%) | Prediction Changes |
|---------|--------------|-----------------|--------------------|
| Dist.TW | 63.31        | 62.59           | 1                  |
| Mid.TW  | 57.79        | 57.79           | 0                  |
| BME     | 77.33        | 80.67           | 3                  |
| Dod.LD  | 31.17        | 32.47           | 2                  |

**Statistical significance of performance gains.** We conduct five repeated trials of ReasonTSC with DeepSeek-R1 and GPT-4o-mini on subset BME. The average accuracy and standard deviation in Table 6 demonstrate that leveraging LLMs on top of TSFM baselines consistently offers added value.

Table 6: Performance comparison with five repeated trials. (%)

|                          | Exp1  | Exp2  | Exp3  | Exp4  | Exp5  | Mean  | std  |
|--------------------------|-------|-------|-------|-------|-------|-------|------|
| MOMENT                   | 74.00 | 75.33 | 77.33 | 70.00 | 76.67 | 74.67 | 2.60 |
| `ReasonTSC` (GPT-4o-mini) | 74.67 | 74.00 | 78.00 | 74.00 | 73.33 | 74.80 | 1.66 |
| `ReasonTSC` (DeepSeek-R1) | 80.67 | 77.33 | 85.33 | 82.67 | 85.33 | 82.27 | 3.03 |

## B.6 GENERALIZABILITY ANALYSIS OF OF `ReasonTSC`

**Anomaly detection task.** The `ReasonTSC` framework achieves robust generalization on anomaly detection tasks. We compare `ReasonTSC` against the AnomLLM across diverse anomaly-specific datasets (Zhou & Yu, 2024). AnomLLM relies on the basic prompt *Let's think step by step*. In contrast, under zero-shot settings, we replace the vanilla chain-of-thought (CoT) with `ReasonTSC`'s tailored step-by-step reasoning strategy, evaluating it on five critical anomaly-type datasets *(range, freq, point, noisy-freq, and noisy-point)* defined by AnomLLM. As shown in Table 7, `ReasonTSC` with GPT-4o-mini outperforms AnomLLM across datasets, including Range, Freq, Noisy-freq, and Noisy-point, demonstrating its stronger generalization and underscoring the value of task-adaptive reasoning in time series anomaly detection.

**Vision-based task.** The `ReasonTSC` framework can be easily generalized to vision-based tasks. VL-Time (Liu et al., 2025a) is a prompt-based approach combining visualized time-series data with vanilla CoT reasoning. We integrate `ReasonTSC`'s tailored step-by-step reasoning in the third round into VL-Time and evaluate it on the EMG dataset under zero-shot settings. Table 8 reveals that `ReasonTSC` substantially enhances LLMs' reasoning capabilities compared to baseline methods.

Table 7: Performance of `ReasonTSC` on anomaly detection reasoning tasks.

| Anomalies | CoT | Precision | Recall | F1 | Affi Precision | Affi Recall | Affi F1 |
|---|---|---|---|---|---|---|---|
| Range | ReasonTSC | 13.02 | 14.29 | 12.39 | 46.69 | 43.71 | 43.94 |
| Range | Anomllm | 10.00 | 10.61 | 9.21 | 31.39 | 30.25 | 30.00 |
| Freq | ReasonTSC | 34.80 | 28.00 | 31.10 | 60.50 | 43.20 | 50.40 |
| Freq | Anomllm | 3.95 | 8.88 | 4.53 | 33.36 | 36.97 | 33.79 |
| Noisy-freq | ReasonTSC | 3.40 | 3.70 | 3.60 | 52.50 | 41.20 | 46.20 |
| Noisy-freq | Anomllm | 2.94 | 4.54 | 2.84 | 25.88 | 24.39 | 24.13 |
| Noisy-point | ReasonTSC | 8.10 | 100.00 | 15.00 | 68.60 | 100.00 | 81.40 |
| Noisy-point | Anomllm | 3.29 | 3.06 | 2.72 | 22.44 | 25.08 | 23.00 |

Table 8: Performance of `ReasonTSC` on vision-based task under zero-shot settings.

| Framework | Model | Accuracy |
|---|---|---|
| VL-Time | GPT-4o | 33.33% |
| VL-Time | Qwen2-VL-72B | 33.33% |
| VL-Time+ReasonTSC | Qwen2.5-VL-3B | 45.00% |
| VL-Time+ReasonTSC | Qwen2.5-VL-7B | 41.46% |

**Multivariate reasoning task.** Given the context length limitations of general LLMs for multivariate UEA datasets, we use <dim> tags to segment different dimensions within each sample, and apply interval sampling on the time series data to mitigate extremely long sequences by concatenating all dimensions. On the Eplp. subset (60 samples, 4 classes, 3 dimensions, 206 time points per dimension) with 1:4 interval sampling, `ReasonTSC` yields accuracies of **80.70%** (DeepSeek-R1) and **73.68%** (GPT-4o-mini), surpassing the plug-in model MOMENT (71.93%). This validates that `ReasonTSC` generalizes to reasoning over multivariate time series and improves accuracy.

### B.7 PERFORMANCES OF REASONTSC WITH MAINSTREAM LLMS

Table 9 presents the complete performance of `ReasonTSC` integrated with *sixteen* mainstream LLMs, as discussed in Subsection 3.2. Notably, `ReasonTSC` substantially enhances DeepSeek-V3's performance in the Mid.OA dataset, improving its accuracy from 27.92% to 66.88% compared to the vanilla CoT approach without plug-in models. Additionally, it enables DeepSeek-V3 to achieve performance comparable to DeepSeek-R1. Besides, gemini-2.5-pro (175B parameters) and claude-3.7-sonnet (120B parameters) demonstrate superior performance owing to their inherent reasoning capabilities acquired during reasoning-enhancing post-training.

### B.8 COST ANALYSIS OF REASONTSC

We measure the inference time and API costs of `ReasonTSC` and Vanilla CoT per sample on the UCR/UEA subsets. As shown in Table 10, while the Vanilla CoT is slightly more efficient, the overall overhead for both methods is comparable because time series data dominates the prompt length. Nevertheless, `ReasonTSC` achieves significantly better performance while also providing explainable rationales..

## C REASONING DETAILS OF OUR PROPOSED REASONTSC FRAMEWORK

This Section presents the full details of the prompt templates devised by `ReasonTSC` in Subsection C.1, followed by the responses of three LLMs in the third reasoning round in Subsection C.2. Subsection C.3 presents the templates of mainstream reasoning techniques used as baselines in this work.

Table 9: The performance of `ReasonTSC` with mainstream LLMs as reasoning language models on three UCR/UEA datasets (Mid.OA, BME and ERing).

| LLM | Parameter | Mid.OA | BME | ERing | Average |
|---|---|---|---|---|---|
| MOMENT (*plug-in*) | 341M | 60.39 | 74.00 | 72.59 | 68.99 |
| Vanilla CoT (gemini-2.5-pro) | 175B | 37.01 | 79.33 | 57.78 | 58.04 |
| Vanilla CoT (deepseek-v3) | 671B | 27.92 | 63.23 | 60.00 | 50.38 |
| gpt-4o-mini | 8B | 61.04 | 77.33 | 74.81 | 71.06 |
| gpt-3.5-turbo | 20B | 59.74 | 74.00 | 72.59 | 68.78 |
| gpt-5-nano | - | 59.09 | 82.00 | 73.33 | 71.47 |
| qwen3-30b-a3b | 30B | 61.69 | 77.18 | 74.07 | 70.98 |
| qwen3-32b | 32B | 59.09 | 77.33 | 72.59 | 69.67 |
| llama-3.3-70b-Instruct | 70B | 61.04 | 84.00 | 74.07 | 73.04 |
| qwen2.5-72B-Instruct | 72B | 60.39 | 78.67 | 72.59 | 70.55 |
| claude-3.7-sonnet | 120B | 62.99 | 82.00 | **77.04** | 74.01 |
| claude-3.5-haiku | 175B | 60.53 | 77.18 | 71.85 | 69.85 |
| gemini-2.0-flash | - | 58.44 | 86.67 | 76.30 | 73.80 |
| gemini-2.5-pro | 175B | 62.34 | 86.00 | 76.30 | **74.88** |
| gpt-4o | 200B | 62.34 | 76.00 | 72.59 | 70.31 |
| qwen3-235b | 235B | 63.64 | 83.33 | 74.07 | 73.68 |
| llama-3.1-405b | 405B | 61.04 | 76.67 | 71.85 | 69.85 |
| deepseek-v3 | 671B | **66.88** | **88.00** | 69.63 | 74.84 |
| deepseek-r1 | 671B | 63.64 | 82.67 | 74.07 | 73.46 |
| grok-3 | - | 64.29 | 81.33 | 76.87 | 74.16 |
| gpt-5 | - | 62.34 | 78.67 | 76.56 | 72.52 |

Table 10: Cost comparison between `ReasonTSC` with GPT-4o-mini and Vanilla CoT.

| Dataset | Time (s) | | Cost | |
|---|---|---|---|---|
| | ReasonTSC | Vanilla CoT | ReasonTSC | Vanilla CoT |
| Dist.TW | 39.53 | 25.35 | 0.0290052 | 0.0174164 |
| Med.Img | 35.05 | 49.65 | 0.0427644 | 0.0294567 |
| BME | 36.79 | 36.36 | 0.0266837 | 0.0171738 |
| Arr.Hd | 79.34 | 40.15 | 0.0404523 | 0.0212289 |
| Dod.LD | 30.63 | 33.28 | 0.0436275 | 0.0261471 |
| Elec. | 30.05 | 28.61 | 0.0345629 | 0.0192602 |

### C.1 FULL PROMPT TEMPLATES FOR THREE-TURN REASONING ROUNDS OF REASONTSC

The proposed `ReasonTSC` develops a multi-turn reasoning approach with a fused decision-making strategy tailored to TSC. The framework consists of three key reasoning stages: (1) TS Pattern Reasoning. `ReasonTSC` guides the LLM to analyze typical patterns across time series categories. (2) Plug-in Model Fusion Reasoning. Predictions and confidence scores from domain-specific time series models are incorporated as in-context examples. (3) Integrative Step-by-step Reasoning. `ReasonTSC` guides the LLM through a structured reasoning process. It evaluates the initial assessment, backtracks to consider alternative hypotheses, and compares their merits before arriving at a final classification. The complete prompt template for this process is presented below.

The domain-specific knowledge incorporated in `ReasonTSC` is derived from the UCR/UEA Archive's documentation, which provides real-world brief descriptions of each dataset's domain along with explanations of category labels.

---

**1st Round Reasoning Prompt: TS Pattern Reasoning**

### Task Description
You are given a time series classification task with the [dataset name] dataset, [domain-specific knowledge of the dataset]. You will be provided with two time series samples from each

---

category of this dataset. Your first task is to analyze and compare the significant pattern differences across these categories.

### Dataset Details
– Categories: [class count]
– Sequence Length: [sample length] time points

### Time Series Samples (2 samples per category):
Category 1:
– Sample 1: [sample for category 1]
– Sample 2: [sample for category 1]
. . .
Category $k$:
– Sample 1: [sample for category $k$]
– Sample 2: [sample for category $k$]

### Analysis Task
Compare and summarize the significant differences in the time series patterns across categories based on the following characteristics. Explicitly state if no differences are observed. Break the series into meaningful segments (e.g. early, middle, late) if applicable.

### Answer Format
– Trend Differences: [Describe trends (upward/downward) and how trends differ across categories, or state if no trends are observed.]
– Cyclic Behavior Differences: [Describe differences in cyclic or periodic patterns, or state if none are found.]
– Stationarity Differences: [Describe stability or shifts in the time series, or state if none are found.]
– Amplitude Differences: [Compare constant or fluctuating amplitudes, or state if no differences]
– Rate of Change Differences: [Describe the speed of change across categories (rapid, moderate, slow), or state if none are found.]
– Outliers Differences: [Identify distinct outliers or anomalies, or state if none are found.]

---

### 2nd Round Reasoning Prompt: Plug-in Model Fusion Reasoning

### Task Description
You are given a time series classification task with the [dataset name] dataset, [domain-specific knowledge of the dataset]. Your second task is to analyze the time series data and refine your understanding based on the classification results and logits (model probabilities for each category) provided by a domain-specific model.

### Dataset Details
– Categories: [class count]
– Sequence Length: [sample length] time points

### Model Details
– Classification Accuracy: [performance of plug-in model (%)]

### Classificaition Examples
– Case 1: True Label: [ground truth], Model Result: [plug-in model's prediction], Category Logits: [plug-in model's logits], Time Series Sample: [time series sample]
– Case 2: True Label: [ground truth], Model Result: [plug-in model's prediction], Category Logits: [plug-in model's logits], Time Series Sample: [time series sample]
– Case 3: True Label: [ground truth], Model Result: [plug-in model's prediction], Category Logits: [plug-in model's logits], Time Series Sample: [time series sample]

### Analysis Task
Refine your understanding of the time series patterns, considering the model's classification results and logits. Identify any necessary adjustments to your initial analysis.

### Answer Format
– Classification Analysis: [Evaluate the logits' confidence and alignment with categories.]
– Time Series Understanding Adjustment: [Adjust your understanding of time series patterns based on the model's results.]

---

**3rd Round Reasoning Prompt: Integrative Step-by-step Reasoning**

### Task Description
Based on your refined understanding, your third task is to perform the time series classification task on the new data sample. You will use your updated analysis of time series patterns along with the result and category logits (model probabilities for each category) from the domain-specific model to make a final classification decision.
### Dataset Details
– Categories: [class count]
– Sequence Length: [sample length] time points
### Model Details
– Classification Accuracy: [accuracy of plug-in model %]
### Classification Task
– Task: Model Result: [plug-in model's prediction], Category Logits: [plug-in model's logits], Time Series Sample: [time series sample]
Please think step by step:
– Analyze the Time Series Pattern: [Examine the time series data for trends, cyclic behavior, stationarity, amplitude, rate of change, and outliers. Compare these characteristics across the categories to identify any significant patterns or differences.]
– Interpret the Model's Results: [Evaluate the model's classification result and logits. Assess the confidence level of the model's prediction and how well it aligns with the observed time series patterns.]
– Make a Preliminary Prediction: [Based on your analysis of the time series pattern and the model's results, make an initial classification decision. Provide a brief explanation for this decision.]
– Review Alternative Classifications: [Consider if there are any other plausible categories that could fit the observed time series pattern. Evaluate the strengths and weaknesses of these alternative classifications compared to your initial prediction.]
– Final Classification Decision: [After reviewing all possibilities, make your final classification decision.]
– True Label: [Your Final Classification Result]

---

C.2    ILLUSTRATIVE RATIONALES GENERATED BY ReasonTSC WITH DEEPSEEK, LLAMA, AND GPT

In this subsection, we demonstrate illustrative generations in the third reasoning round from ReasonTSC integrated with DeepSeek, Llama, and GPT. Specifically, these illustrative generations cover three representative cases: (1) ReasonTSC with DeepSeek identifies and corrects the plug-in model's biased prediction by analyzing its behavioral tendency; (2) ReasonTSC with Llama initially agrees with the plug-in model's prediction but subsequently overrides it after detecting closer logit values and more representative temporal patterns in category 6; (3) ReasonTSC with GPT maintains consistency with the plug-in model's final prediction after analysis of temporal characteristics and the category-wise logit distributions.

---

**ReasonTSC with DeepSeek identifies and corrects the plug-in model's biased prediction by analyzing its behavioral tendency.**

Ground Truth = 6, Plug-in Model Prediction = 4, LLM Prediction = 6
==========================**Analyze the Time Series Pattern**==========================
– Spike Timing: The significant spike (9.134) occurs at positions 15–23 (mid-series), characteristic of Category 6* (initial analysis showed mid-series spikes for Cat 6 vs. late spikes for Cat 4). – Amplitude: The spike magnitude (9.134) aligns with Category 6's highest observed outlier (9.13 in initial samples). – Post-Spike Behavior: Smaller oscillations (1.156, 0.745, etc.) resemble Category 6's gradual decline after a spike.
==========================**Interpret the Model's Results**==========================

---

– The model predicts Category 4 (logit=2.0) but shows moderate support for Category 6 (logit=0.98). This mirrors previous misclassifications (e.g., Case 3) where the model prioritized amplitude over spike timing.

========================= **Make a Preliminary Prediction** =========================

– The mid-series spike strongly matches Category 6's defining pattern. The model's bias toward Category 4 likely stems from overemphasizing spike magnitude rather than temporal position.

=========================**Review Alternative Classifications**=========================

– Category 4: Plausible due to high spike amplitude, but its defining late-spike pattern (e.g., positions 80–90 in initial samples) is absent here. – Category 3: Rejected due to lack of irregular scattered spikes. – Category 1/7: No sustained flat baseline or post-spike oscillations.

========================= **Final Classification Decision** =========================

– Despite the model's output, the spike timing and post-spike dynamics align unambiguously with Category 6.

---

`ReasonTSC` with Llama initially aligns with the plug-in model's prediction but subsequently overrides it after detecting category 6's closer logit values and more representative temporal patterns.

Ground Truth = 6, Plug-in Model Prediction = 3, LLM Prediction = 6

=========================**Analyze the Time Series Pattern**=========================

The given time series data shows a constant trend until around the 50th time point, where it significantly increases (to 9.607) and then decreases back to the baseline level. This pattern is indicative of a specific event or anomaly. The amplitude of the time series is generally low except for the mentioned spike. The rate of change is moderate, with a rapid increase and decrease around the 50th time point. There are no clear cyclic patterns, and the series appears to be stationary except for the outlier.

========================= **Interpret the Model's Results** =========================

The model predicts Category 3 with a logit of 1.02, which is the highest among all categories but not significantly higher than Category 6 (logit of 0.78) or Category 4 (logit of 0.81). This suggests that while the model has some confidence in its prediction, there is ambiguity, and other categories could also be plausible.

========================= **Make a Preliminary Prediction** =========================

Based on the time series pattern and the model's results, my initial classification decision is Category 3. The reasoning behind this decision is the model's prediction and the presence of a significant spike in the time series, which might align with patterns observed in Category 3. However, the confidence in this prediction is not high due to the model's ambiguity and the similarity of the time series pattern to patterns that could be observed in other categories.

=========================**Review Alternative Classifications**=========================

Considering the time series pattern, Category 6 could also be a plausible classification. The pattern of a significant increase followed by a return to baseline could fit the characteristics of Category 6, as observed in previous analyses. Additionally, the model's logit for Category 6 (0.78) is not significantly lower than that of Category 3, suggesting that the model considers Category 6 as a possible classification as well.

========================= **Final Classification Decision** =========================

After reviewing the time series pattern, the model's results, and considering alternative classifications, my final classification decision is Category 6. The reason for choosing Category 6 over Category 3 is the closer alignment of the time series pattern with the characteristics observed in Category 6, such as the initial increase followed by a decrease back to the baseline level. Although the model's prediction was Category 3, the ambiguity in the model's results and the closer match of the time series pattern to Category 6 characteristics lead to this final decision.

> `ReasonTSC` with GPT maintains consistency with the plug-in model's final prediction after analysis of temporal characteristics and the category-wise logit distributions.

Ground Truth = 1, Plug-in Model Prediction = 1, LLM Prediction = 1

========================= **Interpret the Model's Results** =========================

The model predicted Category 1. The logits are [2.62, -1.15, -1.37]. The positive logit for Category 1 (2.62) indicates strong confidence in this classification, while the negative logits for Categories 2 and 3 suggest that the model sees these categories as unlikely given the observed time series.

========================= **Make a Preliminary Prediction** =========================

Based on the analysis of the time series pattern, which shows high fluctuations, rapid changes, and cyclic behavior consistent with Category 1, along with the model's strong confidence in this classification, I would initially classify this time series as Category 1.

=========================**Review Alternative Classifications**=========================

Category 2: This category is characterized by more stable and linear patterns. Given the high fluctuations and rapid changes in the observed time series, Category 2 does not seem plausible. Category 3: While there are some similarities in trends, Category 3 is generally more stable with less pronounced fluctuations. The model's logits indicate that Category 3 is unlikely, as evidenced by the negative logit value. Strengths and Weaknesses: Strengths of Category 1: The observed time series aligns well with the characteristics of Category 1, and the model's logits support this classification. Weaknesses of Category 3: Although there are some overlapping patterns, the overall behavior of the time series is more consistent with Category 1.

========================= **Final Classification Decision** =========================

After reviewing all possibilities and considering the strong alignment of the time series with the characteristics of Category 1, I confirm the classification as Category 1.

## C.3   PROMPT TEMPLATES FOR MAINSTREAM REASONING BASELINE TECHNIQUES

We present the template of the Vanilla Chain-of-Thought technique used in Tables 1 and 2, as well as the Self-Consistency technique used in Figure 2. For the Self-Correction method, the initial prompt is the same as Self-Consistency. In the second and third reasoning rounds, we provide the LLM with its prior prediction and append the instruction *Please review and improve the classification result*.

> ### Vanilla Chain-of-Thought
>
> You are given a time series classification task with the [dataset name] dataset, [domain-specific knowledge of the dataset]. You will be provided with two time series samples from each category of this dataset.
> **### Dataset Details**
> – Categories: [class count]
> – Sequence Length: [sample length] time points
> **### Time Series Samples (2 samples per category):**
> Category 1:
> – Sample 1: [sample for category 1]
> – Sample 2: [sample for category 1]
> . . .
> Category $k$:
> – Sample 1: [sample for category $k$]
> – Sample 2: [sample for category $k$]
> You will be give 3 samples accompanied by the prediction and category logits (model probabilities for each category) from the domain-specific model.
> **### Classificaition Examples**
> – Case 1: True Label: [ground truth], Model Result: [plug-in model's prediction], Category Logits: [plug-in model's logits], Time Series Sample: [time series sample]
> – Case 2: True Label: [ground truth], Model Result: [plug-in model's prediction], Category Logits: [plug-in model's logits], Time Series Sample: [time series sample]

– Case 3: True Label: [ground truth], Model Result: [plug-in model's prediction], Category Logits: [plug-in model's logits], Time Series Sample: [time series sample]
Based on your understanding, now you need to perform the time series classification task on the new data sample.
– Time Series Sample: [time series sample]
**Please think step by step.**
**### Answer Format**
– Explanation: [Brief explanation for your classification decision.]
– True Label: [Your Final Classification Result]

Self-Consistency: the technique involves repeating the prompt three times and selecting the most consistent result.

You are given a time series classification task with the [dataset name] dataset, [domain-specific knowledge of the dataset]. You will be provided with two time series samples from each category of this dataset.
**### Dataset Details**
– Categories: [class count]
– Sequence Length: [sample length] time points
**### Time Series Samples (2 samples per category):**
Category 1:
– Sample 1: [sample for category 1]
– Sample 2: [sample for category 1]
. . .
Category $k$:
– Sample 1: [sample for category $k$]
– Sample 2: [sample for category $k$]
You will be give 3 samples accompanied by the prediction and category logits (model probabilities for each category) from the domain-specific model.
**### Classification Task:**
Classify the following time series data into one of the provided categories based on the patterns observed in the samples.
– Time Series Sample: [time series sample]
**### Answer Format**
– Explanation: [Brief explanation for your classification decision.]
– True Label: [Your Final Classification Result]

# D   IMPLEMENTATION DETAILS OF OUR PROPOSED REASONTSC

This section presents the implementation details of ReasonTSC. Specifically, Subsection D.1 describes the representative datasets selected from the UCR/UEA archive. Subsection D.2 describes the mainstream LLMs employed for evaluating ReasonTSC, and Subsection D.3 introduces existing time series foundation models.

## D.1   DATASET DETAILS OF UCR/UEA ARCHIVE

We evaluate ReasonTSC on 15 representative time series datasets, 9 from the UCR archive and 6 from the UEA archive. These datasets cover diverse domains and vary in key characteristics (e.g., number of classes, time series length) to assess the performance stability of ReasonTSC (Subsection B.5). Detailed information about these datasets is summarized in Table 11.

## D.2   DETAILS OF ADOPTED LLMS

This subsection introduces the mainstream LLMs evaluated in our study. We examine six representative model series: Gemini, Llama3, GPT, DeepSeek, and Grok3, which span diverse parameter scales from 8B (GPT-4o-mini) to 671B (DeepSeek). To assess the impact of ReasonTSC's TSC-tailored

Table 11: The dataset details of the UCR/UEA Archive.

| Dataset | Type | Train Size | Test Size | Classes | Length | Domain |
|---|---|---|---|---|---|---|
| DistalPhalanxTW | IMAGE | 400 | 139 | 6 | 80 | Medical |
| MiddlePhalanxTW | IMAGE | 399 | 154 | 6 | 80 | Medical |
| MiddlePhalanxOutline AgeGroup | IMAGE | 400 | 154 | 3 | 80 | Medical |
| MedicalImages | IMAGE | 381 | 760 | 10 | 99 | Medical |
| ElectricDevices | DEVICE | 8926 | 7711 | 7 | 96 | Energy |
| BME | SIMULATED | 30 | 150 | 3 | 128 | Shape |
| ArrowHead | IMAGE | 36 | 175 | 3 | 251 | Cultural |
| DodgerLoopDay | SENSOR | 78 | 80 | 7 | 288 | Traffic |
| CBF | SIMULATED | 30 | 900 | 3 | 128 | Shape |
| RacketSports | HAR | 151 | 152 | 4 | 30 | Sports |
| ERing | HAR | 30 | 270 | 6 | 65 | Gesture |
| NATOPS | HAR | 180 | 180 | 6 | 51 | Gesture |
| Libras | HAR | 180 | 180 | 15 | 45 | Gesture |
| Epilepsy | HAR | 137 | 138 | 4 | 207 | Medical |
| PenDigits | MOTION | 7494 | 3498 | 10 | 8 | Handwriting |

CoT, we categorize these models into two groups: those enhanced with reasoning training techniques and those without. It is worth noting that powerful reasoning LLMs such as GPT-o1 are excluded from evaluation due to high per-token pricing. However, based on experimental results from the 16 evaluated LLMs, GPT-o1 has the potential to achieve comparable or even superior classification accuracy when integrated with `ReasonTSC`. An overview of these LLMs is provided in Table 12.

Table 12: Overview of the sixteen mainstream LLMs integrated with `ReasonTSC` for evaluation.

| LLM | Parameters | Reasoning-enhancing post-training | Developer | Release |
|---|---|---|---|---|
| Gemini-2.5-pro | 175B | ✓ | Google | 2025-03 |
| Gemini-2.0-flash | Unknown | ✓ | Google | 2024-12 |
| Llama-3.3-70b-instruct | 70B | ✗ | Meta | 2024-12 |
| Llama-3.1-405b-instruct | 405B | ✗ | Meta | 2024-07 |
| GPT-4o-mini | 8B | ✗ | OpenAI | 2024-07 |
| GPT-4o | 200B | ✗ | OpenAI | 2024-11 |
| GPT-3.5-turbo | 20B | ✗ | OpenAI | 2024-01 |
| GPT-5-nano | Unknown | ✓ | OpenAI | 2025-08 |
| GPT-5 | Unknown | ✓ | OpenAI | 2025-08 |
| DeepSeek-V3 | 671B | ✗ | DeepSeek | 2024-12 |
| DeepSeek-R1 | 671B | ✓ | DeepSeek | 2024-01 |
| Grok3 | Unknown | ✗ | xAI | 2025-02 |
| Claude-3.7-sonnet | 120B | ✓ | Anthropic | 2025-02 |
| Claude-3.5-haiku | 175B | ✗ | Anthropic | 2024-10 |
| Qwen3-235b-a22b | 235B | ✓ | Alibaba | 2025-04 |
| Qwen3-30b-a3b | 30B | ✓ | Alibaba | 2025-04 |
| Qwen3-32b | 32B | ✓ | Alibaba | 2025-04 |
| Qwen2.5-72B-Instruct | 72B | ✗ | Alibaba | 2024-09 |

### D.3 DETAILS OF ADOPTED TIME-SERIES FOUNDATION MODELS

In this subsection, we briefly introduce the current time series foundation models (TSFMs), with detailed comparisons provided in Table 13. Our analysis reveals three observations: (1) TSFMs generally adopt large language models as their backbone and primarily process single-modality time series data for both input and output, lacking natural language interaction capabilities. (2) Most TSFMs focus on time series forecasting, while tasks such as classification, anomaly detection and imputation remain unexplored. (3) The reasoning ability of TSFMs are largely unexplored. Current research has not adequately determined whether the success of TSFMs stems from memorizing training patterns or genuine reasoning abilities

Transformer architectures have demonstrated state-of-the-art performance in diverse time series tasks (Liu et al., 2024f; Wang et al., 2024c). Inspired by the success of pretrained LLMs, researchers have shifted toward developing time series foundation models (TSFMs) with zero-shot or few-shot capabilities (Liang et al., 2024), which generally fall into three architectural categories: encoder-only, decoder-only, and encoder-decoder structures. For instance, MOMENT (Goswami et al., 2024), an encoder-only TSFM based on T5, supports diverse time series tasks such as forecasting, classification and anomaly detection. Similarly, MOIRAI (Woo et al., 2024) is a masked encoder-based universal time series forecaster designed for zero-shot tasks. In contrast, decoder-only models possess the ability for iterative generation. Given sequential input patches, they autoregressively predict the next patch conditioned on all preceding ones. Representative examples include TimesFM (Das et al., 2024), featuring a decoder-style attention architecture with input patching, and Timer (Liu et al., 2024h), which leverages an autoregressive approach for generative pre-training. Besides, Chronos (Ansari et al., 2024) primarily focus on the variants of the encoder-decoder T5 model, and uses simple scaling and quantization to tokenize time series into discrete bins.

Given that pre-trained TSFMs incur substantial computational costs, some studies leverage language models pre-trained on billions of tokens for time series analysis. GPT4TS (Zhou et al., 2023) and LLM4TS (Chang et al., 2025) freeze the pretrained blocks and fine-tune positional embeddings and layer normalization, achieving comparable performance in time series tasks. Time-LLM (Jin et al., 2024a) introduces a reprogramming framework that bridges the modality gap between time series data and natural language. Similarly, UniTime (Liu et al., 2024d) is a cross-domain learning approach that utilizes domain instructions and a Language-TS Transformer to provide identification information. Additionally, TEMPO (Cao et al., 2024) introduces an interpretable, prompt-tuning-based GPT architecture to focus on leveraging knowledge from distinct temporal semantics components.

Table 13: Overview of time series foundation models.

| Model | Base Architecture | Text as Input | Text Generation | Forecasting | Classification | Anomaly Detection | TS Reasoning |
|---|---|---|---|---|---|---|---|
| MOMENT | T5 encoder | ✗ | ✗ | ✓ | ✓ | ✓ | ✗ |
| Chronos | T5 (encoder-decoder) | ✗ | ✗ | ✓ | ✗ | ✗ | ✗ |
| MOIRAI | Encoder-only Transformer | ✗ | ✗ | ✓ | ✗ | ✗ | ✗ |
| TimesFM | Decoder-only | ✗ | ✗ | ✓ | ✗ | ✗ | ✗ |
| Timer | Decoder-only | ✗ | ✗ | ✓ | ✗ | ✓ | ✗ |
| GPT4TS | GPT2 | ✗ | ✗ | ✓ | ✓ | ✓ | ✗ |
| LLM4TS | GPT2 | ✗ | ✗ | ✓ | ✗ | ✗ | ✗ |
| Time-LLM | Llama-7B | ✓ | ✗ | ✓ | ✗ | ✗ | ✗ |
| UniTime | GPT2 | ✓ | ✗ | ✓ | ✗ | ✗ | ✗ |
| TEMPO | GPT2 | ✗ | ✗ | ✓ | ✗ | ✗ | ✗ |

# E   INTERPRETATION STUDY – CAN REASONTSC REASON ABOUT TS PATTERNS?

This section presents a comprehensive evaluation of how our proposed ReasonTSC reasons about TS patterns as discussed in Subsection 3.4.1. In the following, Subsection E.1 evaluates ReasonTSC's ability to think over TS patterns with four synthetic datasets; Subsection E.2 conducts this investigation with the real-world UCR/UEA datasets. Additionally, Subsection E.3 provides illustrative rationales generated by ReasonTSC integrated with two mainstream LLMs.

## E.1   EVALUATION ON SYNTHETIC DATASETS

**Synthetic Datasets Generation.** To evaluate ReasonTSC's ability to interpret general TS patterns. For illustration, we generate four synthetic datasets, each containing 200 TS samples with one of the four pattern types (i.e., *Trend, Frequency, Amplitude, and Mixed Patterns*) with a fixed sequence length of 100. Following (Cai et al., 2024; Potosnak et al., 2024), we devise a series of particular linear functions to simulate time series with the desired patterns. To be specific, the *trend* pattern is simulated by linear plots that exhibit obvious slopes $y(t) = \beta_0 + \beta_1 \cdot t + \epsilon(t)$; the *frequency* pattern is simulated by sine functions with fixed frequencies and amplitudes $y(t) = A \cdot \sin(2\pi f \cdot t + \phi) + \epsilon(t)$; the *amplitude* pattern is reflected by sine functions with varying amplitudes $y(t) = A \cdot \sin(2\pi f \cdot t) + \epsilon(t)$; the *mixed patterns* pattern is captured by integrating multiple sine functions with varying frequencies and

phases to illustrate cases where various complex TS patterns are presented within adjacent samples $y(t) = \beta_0 + \beta_1 \cdot t + A_1 \cdot \sin(2\pi f_1 \cdot t + \phi_1) + A_2 \cdot \cos(2\pi f_2 \cdot t + \phi_2) + \epsilon(t)$. Besides, we also construct plain time series samples without distinguishable features compared with the aforementioned patterns by introducing Gaussian noise to the time series, serving as negative counterparts.

**Evaluation Process.** We present each time-series sample in a multiple-choice format alongside a randomly generated noise sequence, prompting the `ReasonTSC` to identify the sequence with the most discernible patterns. The choice positions are randomized to eliminate positional bias. The prompt templates are illustrated below.

---

#### Trend

Compare the two provided time series samples and select the one that demonstrates a more typical and well-defined trend pattern, specifically a sustained and clear directional trend (either upward or downward) throughout the series.
– Case A: [time series sample]
– Case B: [time series sample]
### Answer Format:
– Option: [Case A or Case B]
– Explanation: [Reason for choosing this time series sample and the specific pattern observed]

---

#### Frequency

Compare the two provided time series samples and select the one that exhibits a more typical and well-defined frequency or cyclical pattern, characterized by consistent and regular periodic behavior or repetitive cycles throughout the series.
– Case A: [time series sample]
– Case B: [time series sample]
### Answer Format:
– Option: [Case A or Case B]
– Explanation: [Reason for choosing this time series sample and the specific pattern observed]

---

#### Amplitude

Compare the two provided time series samples and select the one that demonstrates a more typical and well-defined amplitude pattern, characterized by consistent and pronounced variations in value range, indicative of strong oscillations or signal intensity.
– Case A: [time series sample]
– Case B: [time series sample]
### Answer Format:
– Option: [Case A or Case B]
– Explanation: [Reason for choosing this time series sample and the specific pattern observed]

---

#### Mixed Patterns

Compare the two provided time series samples and select the one that exhibits more typical and well-defined patterns, such as trends, seasonality, or cyclical behavior.
– Case A: [time series sample]
– Case B: [time series sample]
### Answer Format:
– Option: [Case A or Case B]
– Explanation: [Reason for choosing this time series sample and the specific pattern observed]

---

**ReasonTSC's Performance of Understanding TS patterns.** Table 14 presents the experimental results. Notably, `ReasonTSC` with GPT, Llama, and Deepseek achieve satisfactory accuracy across all the tested datasets, demonstrating ReasonTSC's ability to generate rationales about fundamental time series patterns.

Table 14: Pattern Recognition Accuracy of LLMs employed in `ReasonTSC` on synthetic data

| LLM | Trend | Frequency | Amplitude | Mixed Patterns |
|---|---|---|---|---|
| GPT-4o-mini | 100% | 100% | 100% | 100% |
| Llama-3.3-70b-instruct | 100% | 100% | 100% | 99.50% |
| DeepSeek-R1 | 100% | 100% | 100% | 100% |

### E.2 EVALUATION ON THE UCR/UEA ARCHIVE

**Data Preparation.** We further employ the UCR/UEA archives for interpretation analysis. For each sample, we randomly select one unique time series instance per category, generating a maximum of 100 samples per dataset. Datasets with fewer than 30 unique samples are excluded.

**Evaluation Process.** In this study, we extend six fundamental time-series patterns to ten: *trend, cyclic, stationarity, amplitude, rate of change, outliers, noise, volatility, structural break, and mean shift* (Cai et al., 2024). We then prompt the `ReasonTSC` to identify significant pattern variations across categories. For each sample, we randomly select one unique instance per category and ask the `ReasonTSC` to identify significant pattern differences across categories. We quantitatively summarize the responses by counting the top three most frequently identified patterns (including ties) and calculating their relative weights. The complete prompt details are provided below.

---

**Pattern Interpretation Prompt Template on the UCR/UEA Archive**

### Task Description
You are given a time series analysis task with the [dataset name] dataset, [domain-specific knowledge of the dataset]. Your task is to analyze and determine whether there are any highly pronounced and distinctly typical temporal patterns across these categories. Only if such patterns are exceptionally clear and consistently representative, mark it as 1; otherwise, mark it as 0.

### Dataset Details
– Categories: [class count]
– Sequence Length: [sample length] time points

### Time Series Samples by Category
– Category 1: [sample for category 1]
– Category 2: [sample for category 2]
. . .
– Category $k$: [sample for category $k$]

### Analysis Task
Compare and summarize the significant differences in the time series patterns across categories based on the following characteristics. Break the series into meaningful segments (e.g. early, middle, late) if applicable. Only mark a characteristic as 1 if the differences are very clear and typical. Explicitly state if no differences are observed.

### Answer Format
– Trend Differences: 0/1. [Describe clear and typical trends (upward/downward) and how they differ across categories, or state if none are found.]
– Cyclic Behavior Differences: 0/1. [Describe clear and typical differences in cyclic or periodic patterns, or state if none are found.]
– Stationarity Differences: 0/1. [Describe clear and typical stability or shifts in the time series, or state if none are found.]
– Amplitude Differences: 0/1. [Describe clear and typical constant or fluctuating amplitudes, or state if none are found.]
– Rate of Change Differences: 0/1. [Describe clear and typical differences in the speed of change (rapid, moderate, slow), or state if none are found.]
– Outliers Differences: 0/1. [Identify clear and typical distinct outliers or anomalies, or state if none are found.]
– Noise Level Differences: 0/1. [Describe clearly and typically the amount of random fluctuations or noise across categories, or state if none are found.]

---

– Volatility Differences: 0/1. [Describe clear and typical differences in variability or fluctuations, or state if none are found.]

– Structural Break Differences: 0/1. [Identify clear and typical significant shifts or breaks in the time series, or state if none are found.]

– Mean Level Differences: 0/1. [Identify clear and typical average values across categories, or state if none are found.]

**ReasonTSC's Performance of Thinking Over TS Patterns.** Table 15 presents our experimental results, with the top three most frequently identified patterns (including ties) highlighted in bold. GPT-4o-mini consistently identifies similar temporal patterns (e.g., trend, amplitude, rate of change, volatility, and mean shift) across all datasets, suggesting that smaller-scale LLMs tend to generate more generalized interpretations. This observation aligns with its performance within the `ReasonTSC` framework. DeepSeek-R1 shows superior capability in identifying category-discriminative patterns. Llama3.3-70b-instruct shows comparable capability, with significant pattern recognition overlap between Llama and DeepSeek, further validating LLMs' temporal reasoning capacities. These observations suggest that an in-depth understanding of the time series patterns could enhance the reasoning process of LLMs and time series classification performance.

Table 15: Pattern Interpretation of LLMs in `ReasonTSC` on UCR/UEA Archive

| Dataset | Samples | LLM | Trend | Cyclic | Stationarity | Amplitude | Rate of Change | Outliers | Noise | Volatility | Structural Break | Mean |
|---|---|---|---|---|---|---|---|---|---|---|---|---|
| BME | 60 | GPT | **60** | 0 | **60** | **60** | **60** | 4 | 37 | **60** | 25 | **60** |
| | | Llama | **60** | 0 | 52 | **60** | **60** | 1 | 0 | 59 | **60** | **60** |
| | | DeepSeek | **53** | 0 | 20 | 37 | 47 | 6 | 2 | 22 | **59** | 48 |
| Arr. Hd | 65 | GPT | **65** | 0 | **65** | **65** | **65** | 0 | 30 | **65** | 15 | **65** |
| | | Llama | 7 | 0 | 1 | **8** | 7 | 0 | 0 | **8** | 2 | **15** |
| | | DeepSeek | 15 | 1 | 3 | **52** | 47 | 2 | 25 | **39** | 8 | 31 |
| CBF | 100 | GPT | **100** | 0 | **100** | **100** | **100** | 76 | 55 | **100** | 61 | **100** |
| | | Llama | 45 | 1 | 3 | **56** | 45 | 10 | 1 | 43 | 41 | **96** |
| | | DeepSeek | 59 | 0 | 23 | **81** | 67 | 16 | 0 | 15 | **95** | **99** |
| Mid. OA | 84 | GPT | **84** | 0 | 83 | **84** | **84** | 0 | 2 | **84** | 40 | **84** |
| | | Llama | **31** | 2 | **31** | **31** | **31** | 1 | 2 | **31** | **31** | 34 |
| | | DeepSeek | 23 | 3 | 32 | 44 | **59** | 32 | 27 | **51** | **59** | 34 |
| Rkt. Spt | 68 | GPT | **68** | 0 | 67 | **68** | **68** | 58 | 10 | **68** | 16 | **68** |
| | | Llama | 42 | 2 | 9 | **68** | 61 | 25 | 4 | **61** | 37 | **68** |
| | | DeepSeek | 50 | 0 | 10 | **56** | **60** | 40 | 27 | **56** | 55 | 49 |
| Eplp. | 60 | GPT | **60** | 0 | **60** | **60** | **60** | **60** | 43 | **60** | 25 | **60** |
| | | Llama | 35 | 2 | 4 | **60** | 43 | 10 | 5 | **58** | 8 | **59** |
| | | DeepSeek | 18 | 56 | 57 | **60** | **60** | 54 | 41 | **60** | 54 | 58 |
| Mid. TW | 34 | GPT | **34** | 0 | **34** | **34** | **34** | 0 | 3 | **34** | 7 | **34** |
| | | Llama | 0 | 0 | 0 | **4** | 0 | 0 | 0 | **2** | 1 | **5** |
| | | DeepSeek | 31 | 1 | 26 | 30 | **34** | 17 | 22 | 31 | **34** | 33 |
| ERing | 50 | GPT | **50** | 0 | **50** | **50** | **50** | 44 | 8 | **50** | 43 | **50** |
| | | Llama | **50** | 0 | 35 | **50** | **50** | 0 | 1 | **50** | **50** | **50** |
| | | DeepSeek | 48 | 0 | 21 | 45 | **50** | 10 | 22 | 45 | **49** | 47 |
| Nt.Ops | 60 | GPT | **60** | 0 | **60** | **60** | **60** | 59 | 2 | **60** | **60** | **60** |
| | | Llama | **60** | 1 | 50 | **60** | **60** | 19 | 2 | **60** | **60** | **60** |
| | | DeepSeek | **59** | 0 | **59** | **59** | **59** | 38 | 7 | **59** | **59** | 56 |
| Elec. | 100 | GPT | 85 | 34 | 84 | **85** | **85** | **85** | 36 | **85** | 48 | **85** |
| | | Llama | 78 | 34 | 73 | **99** | 97 | 98 | 28 | **98** | 86 | **99** |
| | | DeepSeek | 15 | 75 | 65 | **99** | 69 | 88 | 56 | **100** | 96 | 94 |
| Pen. | 100 | GPT | **99** | 0 | 98 | **99** | **99** | 97 | 0 | **99** | 25 | **99** |
| | | Llama | 10 | 8 | 8 | **94** | 87 | 8 | 8 | 61 | 9 | **90** |
| | | DeepSeek | **93** | 0 | 9 | 12 | **94** | 1 | 46 | 90 | 70 | **98** |

## E.3 ILLUSTRATIVE RATIONALES ABOUT TS PATTERNS BY ReasonTSC ON THE UCR/UEA ARCHIVE

We further provide illustrative rationales for time-series patterns generated by Llama-3-70b-instruct and DeepSeek-R1 on the ElectricDevices (Figure 17) and ERing datasets (Figure 18), respectively.

---

**Time-series pattern rationales of Llama-3.3-70b-instruct on the ElectricDevices dataset**

To analyze the time series patterns across the given categories, we'll break down the characteristics as requested. – **Trend Differences**: 0. Upon examining the series, no clear and consistent upward or downward trends are observed across all categories that distinctly differentiate them. – **Cyclic behavior Differences**: 0. There's no evident cyclic or periodic behavior that clearly and typically differentiates the categories. – **Stationarity Differences**: 0. The series generally appear to be non-stationary due to the presence of spikes and shifts, but the differences in stationarity are not clear and typical across categories. – **Amplitude Differences**: 1. There are clear differences in amplitude across categories. Category 4 has a very low amplitude until a significant spike, whereas Category 6 shows a mix of low and high amplitude values. – **Rate of change Differences**: 1. The speed of change varies significantly across categories. For instance, Category 1 remains almost constant, while Category 5 shows rapid changes between two distinct levels. Category 7 has segments with slow changes followed by rapid spikes. – **Outliers Differences**: 1. Outliers or anomalies are present and differ across categories. For example, Category 3 has a significant outlier at the 25th position, and Category 6 has a large spike. – **Noise level Differences**: 0. While there are fluctuations, the level of noise does not clearly and typically differentiate the categories. – **Volatility Differences**: 1. Volatility, or the variability in the series, differs across categories. Category 1 is very stable, whereas Category 5 shows high variability between its two main states. – **Structural Break Differences**: 1. Significant shifts or breaks in the time series are observed and differ across categories. For example, Category 4 has a structural break with a large spike after a long period of stability. – **Mean Level Differences**: 1. The average values across categories differ significantly. For instance, Category 1 has a mean close to -0.179, while Category 6 has a higher mean due to its positive spikes. Category 7's mean is influenced by its mix of negative and positive values.

---

**Time-series pattern rationales of DeepSeek-R1 on the ERing dataset**

– **Trend Differences**: 1. Categories exhibit distinct trend patterns. For example, Category 1 shows a V-shaped trend (sharp decline followed by recovery), Category 4 has a single pronounced peak and trough, and Category 5 displays a gradual decline followed by a sharp drop. – **Cyclic behavior Differences**: 0. No consistent cyclic or periodic patterns (e.g., repeated oscillations) are observed across categories. While Category 3 has multiple abrupt shifts, these are not cyclical but rather irregular structural breaks. – **Stationarity Differences**: 0. All categories exhibit non-stationary behavior (e.g., shifts in mean/variance). – **Amplitude Differences**: 1. Amplitude ranges vary significantly. For instance, Category 2 spans 3.8 (from 1.191 to -1.89), while Category 4 spans 2.0 (from -0.759 to 1.285). These differences in magnitude are pronounced and category-specific. – **Rate of change Differences**: 1. Categories differ in how abruptly values change. Category 1 and Category 4 have rapid drops, Category 5 shows a slow initial decline followed by a sharp drop, and Category 3 has frequent abrupt shifts. – **Outliers Differences**: 0. No distinct outliers are observed; abrupt changes align with structural breaks rather than anomalies. – **Noise level Differences**: 0. All categories exhibit low noise, with values often plateauing for multiple time points. Random fluctuations are minimal and consistent across categories. – **Volatility Differences**: 1. Variability differs: Category 3 and Category 2 show high volatility (frequent large shifts), while Category 6 and Category 4 have stable segments followed by single sharp changes. – **Structural Break Differences**: 1. Structural breaks (sudden shifts) are prominent but differ in timing and frequency. For example, Category 3 has multiple breaks, while Category 6 has one dominant drop. – **Mean Level Differences**: 1. Mean values differ clearly. Categories like Category 1 and 6 start at higher values ( 2.69 and 1.085), while Category 3 and 5 spend significant time at lower means (e.g., -1.384 in Category 5).

## F  LIMITATIONS AND DISCUSSIONS

`ReasonTSC` is a novel framework designed to effectively leverage LLM reasoning for time series classification through a multi-turn tailored reasoning and fused decision-making strategy. However,

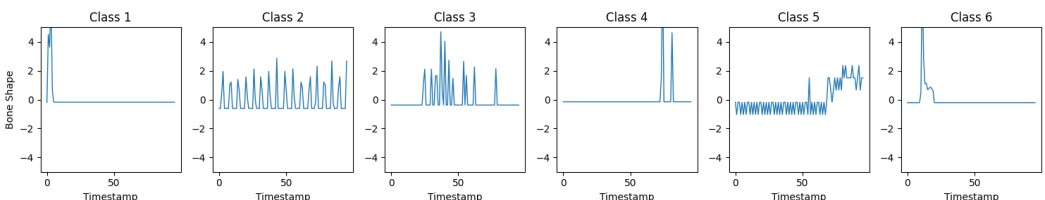

Figure 17: Visualization of Class Distribution in the ElectricDevices Dataset.

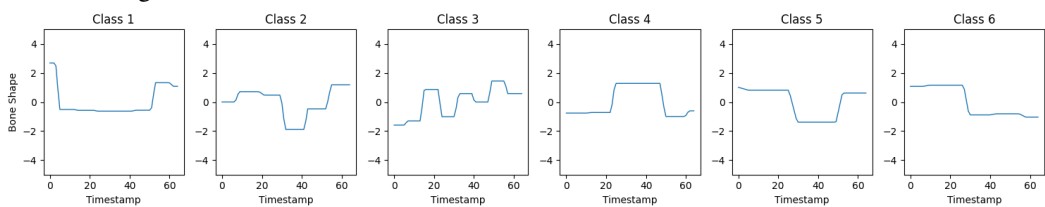

Figure 18: Visualization of Class Distribution in the ERing Dataset.

we also recognize several limitations of this work. First, unlike NLP and vision datasets, time series data typically consists of long sequences with sparse semantics and inherent noise. Due to the context length limits of LLMs, effectively conveying the original time series sequence within a multi-turn prompting framework remains challenging. Second, although the proposed `ReasonTSC` is more cost-efficient than pretraining a time series LLM from scratch, calling APIs of reasoning LLMs still incurs computational overhead. Third, our experiments adhere to the original training-test splits of the UCR/UEA archive. In contrast, different dataset split ratios may impact the performance of plug-in models, which could also indirectly influence the performance of `ReasonTSC`. Last, `ReasonTSC` includes the most fundamental time series patterns in the reasoning process. It is a promising future direction to further exploit domain-specific patterns during the reasoning process to improve domain-specific tasks.

## G   THE USE OF LARGE LANGUAGE MODELS

In this work, large language models (LLMs) are used to polish the writing of the manuscript. Furthermore, since `ReasonTSC` is a novel framework designed to leverage LLM reasoning for time series classification, we evaluate 18 mainstream LLMs to validate its effectiveness. We confirm that the LLMs are not involved in research ideation, experimental design, or results analysis, and take full responsibility for all content and the entire work.

