# OpenReview forum: "Enhancing LLM Reasoning for Time Series Classification by Tailored Thinking and Fused Decision"
_ICLR.cc/2026/Conference — ICLR 2026 Conference Withdrawn Submission_

### Official Review · Reviewer_JuQM · 2025-10-28

**Soundness:** 3
**Presentation:** 3
**Contribution:** 2
**Rating:** 4
**Confidence:** 3

**Summary:**

This paper proposes ReasonTSC which is a framework to facilitate LLM reasoning on time series classification. The reasoning process is segmented into three stages, time series pattern reasoning, plug-in model decision fusion, and lastly assembling all previous information together with backtrack and final decision. Experiments over three distinct LLMs show that ReasonTSC improve over naive CoT reasoning and establishes the validity of LLM time series reasoning.

**Strengths:**

The paper is easy to follow and well presented. The research topic is well positioned among other related works.

Extensive exp and ablation studies demonstrate the effectiveness of ReasonTSC framework.

**Weaknesses:**

Given the multi-turn reasoning setup, I think it's extremely important to provide cost tradeoff experiments and discussions. It's obvious that ReasonTSC framework will be much more expensive then naive CoT inference, but by how much? That we don't know.

The focus on time series classification is rather narrow because this framework won't be able to extend to other time series tasks such as forecasting because you won't be able to backtrack to alternative answers.

**Questions:**

In line 303, it says "we use the first dimension of the multivariate UEA datasets to address the token limit". Does that mean you only used the first channel of multivariate time series? That changes the problem to univariate time series classification problem. And if so, what if the distinguishing pattern is actually not present in the first channel? I might understood it wrong.

In figure 4, it seems like more shots actually hurt the performance (e.g. 5-shot is universally poorer than 2-shot across the three datasets(, any hypothesis on this?

I'm not quite sure what figure 6 is meant to show.

In figure 8, how is the override rate and override accuracy calculated for w/o plug in model? If without plugin model, then shouldn't there be no overrides?

---

> ### Author Response · Authors · 2025-11-21
>
> > **W1: Concerns for cost**
>
> Thank you for highlighting this point. We meaure the per-sample inference time and API costs of ReasonTSC and Vanilla CoT (please see Appendix B.8). As expected, Vanilla CoT is slightly more efficient. However, the overall overhead remains comparable because the prompt length is dominated by the time-series input rather than the reasoning turns. As a result, ReasonTSC provides substantially better accuracy while keeping the additional cost within a tolerable range.
>
> |  | Time(s) | Time(s) | Cost |Cost |
> |----------|---------|--------|--------|--------|
> | Dataset | ReasonTSC | Vanilla CoT | ReasonTSC | Vanilla CoT|
> |Dist.TW|39.53|25.35|0.029|0.017|
> |Med.Img|35.05|49.65|0.043|0.029|
> |BME|36.79|36.36|0.027|0.017|
> |Arr.Hd|79.34|40.15|0.040|0.021|
> |Dod.LD|30.63|33.28|0.044|0.026|
> |Elec.|30.05|28.61|0.035|0.019|
>
>
> > **W2: Concerns for framework generalization**
>
> We appreciate your concern. In ReasonTSC, the backtrack step [1] is not limited to enumerating alternative labels. Its primary purpose is to guide the model to self-reflect [2] on its initial reasoning, reassess potential flaws, and refine its final decision. While forecasting tasks do not naturally admit 'alternative answers' in the same discrete sense as classification, the self-reflection mechanism may still be applicable: the model can revisit its preliminary forecast, and correct implausible trends or assumptions.
>
> The generalization of ReasonTSC has been examined beyond classification, including anomaly detection and vision-based time series tasks (please see Appendix B.6 for more details). These results suggest that ReasonTSC’s integrative step-by-step reasoning offers consistent benefits over vanilla CoT. Although further investigation on forecasting is certainly valuable, current experiments indicates that the framework is not inherently limited to classification and can extend to other time-series tasks.
>
> [1] Towards mitigating LLM hallucination via self reflection//Findings of the Association for Computational Linguistics: EMNLP 2023.
>
> [2] LLMs cannot find reasoning errors, but can correct them given the error location//Findings of the Association for Computational Linguistics: ACL 2024.
>
>
> > **Q1: Concerns for multivariate dataset**
>
> In the main results, we use the first dimension of the multivariate datasets to ensure a fair comparison under LLMs' context-length limitations (for some datasets, length × channels × classes would exceed the allowable context). Even with one single dimension, ReasonTSC still outperforms plug-in models. We appreciate your point for cross-variable dependencies are crucial for multivariate time series. We additionally extend ReasonTSC to the multivariate setting by:
>
> ```
> - We use <dim> tags to separate different dimensions within each sample;
>
> - We employ interval sampling on the time series data to mitigate extremely long sequences by concatenating all dimensions.;
>
> - We adapt the prompt to guide the LLM in comparing dimensions separately across categories in the first round of pattern reasoning.
> ```
> On Epilp. dataset (4 classes, 3 dimensions, 206 time points per dimension), using 1:4 interval sampling, ReasonTSC achieves 80.70% (DeepSeek-R1) and 73.68% (GPT-4o-mini), both exceeding the plug-in model MOMENT (71.93%). This suggests that ReasonTSC remains effective when reasoning over multivariate time series.
>
>
> > **Q2: Hypothesis of few-shot performance**
>
> In Figure 4, the performance of ReasonTSC remains generally stable, with a slight drop as the number of shots increases. In our few-shot setting, the number of examples grows proportionally with the number of classes (e.g., a 5-class subset yields 5 examples in the 1-shot case and 25 in the 5-shot case). We hypothesize that this degradation arises from information overload in long prompt-based inputs: as more time-series samples are added, the model may struggle to effectively utilize all information, leading to diminished gains from additional shots.
>
> > **Q3: Explanation for Figure 6**
>
> Figure 6 is designed to illustrate the effectiveness ReasonTSC's first-round TS pattern reasoning. For each dataset, we randomly select one unique time series instance per category, then prompt the LLM to identify significant pattern differences across categories. We quantitatively summarize the responses by counting the top three most frequently identified patterns (including ties).
> Please refer to Appendix E.2 for full evalation process and results. The figure shows that ReasonTSC with GPT consistently identifies similar TS patterns, while DeepSeek (which also shows the best overall classification performance) shows superior performance in identifying category-discriminative pattern. These observations indicate that a better understanding of the time series patterns could enhance the reasoning process of LLMs and the performance accordingly.

---

> > ### Author Response · Authors · 2025-11-21
> >
> > > **Q3: Explanation for Figure 8**
> >
> > Thank you for raising this point. In the w/o plug-in model setting, there are indeed no actual overrides. In Figure 8, the reported 'override rate' and 'override accuracy' are computed post hoc by comparing the LLM's predictions with the plug-in model's outputs, as we still have access to the plug-in model results for analysis. These metrics therefore reflect the disagreement between the two models. The intention of Figure 8 is to highlight the contribution of ReasonTSC's fused decision strategy. We have revised the description in lines 461–463 to clarify this intention in the updated manuscript.

---

> > ### Comment · Reviewer_JuQM · 2025-11-26
> >
> > Thank you for the response! Could you elaborate more on why "prompt length is dominated by the time-series input rather than the reasoning turns"? I assume that in a multi-turn setting, all history are used as input for next turn which will dominate the prompt length. For Q1, just to clarify, in the original submission, using first dimension indeed means just taking the first channel of the time series dataset? And your plug-in model is actually consuming all channels instead?
> >
> > My questions regarding Q2,3,4 have been resolved.

---

> > > ### Author Response · Authors · 2025-11-27
> > >
> > > Thank you for your insightful comments. We acknowledge that our previous phrasing was not sufficiently clear. You are correct that in a multi-turn setting, the entire dialogue history is included in subsequent turns and therefore contributes substantially to the prompt length. In our setup, the context consists of text tokens (e.g., instructions and reasoning) and time series (TS) tokens. Across all datasets, we observe that text tokens account for a small portion of the total, whereas TS tokens dominate the token budget. In the first round, both ReasonTSC and vanilla CoT include 2 randomly selected time series samples for each category, while the text portion of tokens remains relatively small and comparable between the two methods. Thus, ReasonTSC achieves better performance without introducing a substantial increase in the proportion of text tokens, keeping the additional amount of tokens within an acceptable range relative to Vanilla CoT.
> > >
> > >
> > > | |Round | Time Series Tokens | Text Tokens |  Text Tokens (%)|
> > > |------|------|-------|-------|-------|
> > > | Dist.TW| Vanilla CoT |  6,251  |  2,699    |29.99% |
> > > | Dist.TW |ReasonTSC |  7,693   |   3,599  |31.71%  |
> > > | Med.Img |Vanilla CoT |12483| 2981    |19.16% |
> > > | Med.Img |ReasonTSC | 14,267 |  3,923   | 21.45% |
> > > | BME |Vanilla CoT | 5,374 |  1,988   | 27.00%|
> > > | BME |ReasonTSC | 7,677 | 3,200 | 29.42% |
> > > | Arr.Hd| Vanilla CoT |  10,547  | 1,988 | 15.86% |
> > > | Arr.Hd| ReasonTSC |  15,067   | 3,258  | 17.78%  |
> > > | Dod.LD |Vanilla CoT | 11,772 | 3,296  | 21.87% |
> > > | Dod.LD |ReasonTSC | 14,122 | 3,653 | 20.55% |
> > > | Elec. |Vanilla CoT | 8,651 | 2,637|23.36%|
> > > | Elec. |ReasonTSC |10,381 | 3,448  | 24.93% |
> > >
> > > Regarding Q1, to ensure fairness, the LLM only receives the first channel of each time-series sample, and the plug-in model is also trained and evaluated under the univariate setting. Both models therefore operate solely on the first channel.

---

### Official Review · Reviewer_Q2KJ · 2025-10-30

**Soundness:** 3
**Presentation:** 3
**Contribution:** 2
**Rating:** 4
**Confidence:** 5

**Summary:**

The paper proposes a pipeline for time series classification named ReasonTSC that aims to activate large language model reasoning through a three stage prompt design and a fusion step that injects outputs from a plug in time series classifier. Stage one guides the model to analyze canonical time series factors such as trend, periodicity, stationarity, amplitude, rate of change, and outliers. Stage two introduces predictions and confidence scores from a separate classifier and asks the language model to recalibrate its understanding. Stage three performs step by step judgment with explicit backtracking across candidate labels. Experiments on a subset of UCR and UEA datasets report gains over a vanilla chain of thought baseline and show cases where the method corrects errors produced by the external classifier. The paper claims generality across many language models and positions the approach as a training free way to improve time series classification.

**Strengths:**

- This paper articulates a clear task formulation for time series classification with an emphasis on structured factor analysis. The prompt design is readable and reproducible. The three stage template can be implemented with minimal engineering.

- Evidence-aware decision protocol that treats external classifier scores as auditable signals and separates evidence from verdict, which enables calibration and error analysis.

- The work highlights interpretability by asking the model to reason about trend and periodicity and related factors rather than returning only a label.

- Training-free use of the language model with a plug-in interface to existing time series models, which lowers integration cost in small pilots.

**Weaknesses:**

1. The paper lacks methodological novelty. The approach is a modular composition of known prompting patterns that include structured chain of thought, reflective backtracking, and in context use of external scores. There is no new learning objective, no new model component, and no principled decision module beyond scripted prompts.

2. The method injects predictions and confidence scores from the external classifier, sometimes with accuracy summaries. In a small data context this acts as a strong prior that can dominate the language model. The work does not isolate which information is essential.

3. Most comparisons focus on a vanilla chain of thought or a small number of reflective variants. There is no systematic evaluation against stronger reasoning frameworks such as tree based search, program of thought, retrieval augmentation with time series facts, or tool use with statistical tests and spectral analysis.

4. The claim of broad effectiveness across many language models is not supported by a single consolidated table with matched budgets. The core results highlight only a few models. Cost, latency, and context length are not reported in a comparable way.

5. Presented evidence that the method can overturn external classifier errors does not prove stronger reasoning ability. In most cases the language model reacts to confidence patterns that are already visible in the injected numbers. Without tools that verify time series properties, the mechanism is evidence reweighting rather than genuine reasoning.

6. Conclusions are stronger than the experimental support. Claims of consistent superiority and wide generality appear without comprehensive cost reporting, without strict information bound controls, and without coverage of multivariate and long horizon regimes.

**Questions:**

1. Can you provide an information boundary study that isolates what the model truly needs, for example only predictions, predictions with confidence, predictions with dataset level accuracy, and predictions with ground truth, and report the impact of each on accuracy and on overfitting risk?

2. How does the method compare against stronger reasoning and tool use baselines under matched token budgets and matched latency, for example tree based search, program of thought, retrieval augmented prompts with time series facts, or statistical and spectral tool calls?

3. What is the scalability profile on multivariate and long horizon settings, including accuracy versus context length curves, accuracy versus number of channels, and a cost and latency breakdown when you avoid rounding and channel reduction?

---

> ### Author Response · Authors · 2025-11-21
>
> > **W1: Novelty concerns**
>
> Thank you for raising this point. We agree that our framework does not introduce a new learning objective or model component. The contribution instead lies in addressing a modality-specific reasoning gap that existing reasoning techniques do not resolve. Time series (TS) differs fundamentally from natural language in structure and semantic organization, and current prompting paradigms have shown minimal gains when directly applied to TS tasks. This raises a methodological question: how to elicit TS-specific reasoning behaviors from LLMs using only inference guidance.
>
> Unlike symbolic or mathematical domains, TS classification lacks a well-established formal logic. Progress therefore depends on identifying the failure modes of LLMs in this modality and designing elicitation strategies that activate domain-relevant reasoning behaviors. Our work makes a step in this direction through:
>
> (1) LLMs often fail to adopt pattern-oriented thinking, while pattern primitives are essential in TS. Turn 1 explicitly steers the model toward pattern-level analysis.
>
> (2) LLMs exhibit hallucinated certainty and rarely reconsider alternatives. Turn 3 introduces a backtracking mechanism to mitigate this issue.
>
> (3) Naïve few-shot prompting does not directly improve performance due to modality mismatch. Our fused decision strategy integrates TSFM outputs to provide a more reliable signal.
>
> ReasonTSC can reliably correct TSFM prediction errors. This suggests that the proposed elicitation structure activates latent TS-specific reasoning abilities that existing prompting or CoT variants do not trigger. Finally, we would like to remark that there are several concurrent submissions to ICLR 2026 [1–5], which collectively indicate that time series reasoning is an emerging and timely research direction. These works explore valuable and complementary perspectives, such as RL-based reasoning enhancement, modality alignment through encoders, and incorporating the vision modality to enrich TS semantics. Our work investigates whether TS-specific reasoning can be reliably elicited through prompting alone, without additional LLM training or resource-intensive model adaptation. Therefore, our approach serves as a distinct and orthogonal contribution that focuses on activating the TS-reasoning capabilities inherent in LLMs, and the presence of these concurrent ICLR submissions does not diminish the significance of our work.
>
> [1] TimeOmni-1: Incentivizing Complex Reasoning with Time Series in Large Language Models
>
> [2] Unlocking the Value of Text: Event-Driven Reasoning and Multi-Level Alignment for Time Series Forecasting
>
> [3] From Images to Signals: Are Large Vision Models Useful for Time Series Analysis?
>
> [4] AnomSeer: Reinforcing Multimodal LLMs to Reason for Time-Series Anomaly Detection
>
> [5] Time Series Forecasting as Reasoning: A Slow-Thinking Approach with Reinforced LLMs
>
>
> > **W2&Q1: Concerns for fused decision strategy**
>
> Thank you for raising this point. To evaluate the fused decision strategy of ReasonTSC, we conduct ablation studies in Section 3.4.2. The plug-in model provides two types of information: (1) category-wise confidence scores (logits), and (2) the final prediction. Since logits encode both the predicted label (via the highest score) and fine-grained confidence across categories, we therefore perform ablations by first removing the logits and then removing all plug-in information. This results in a clear degradation in performance across all three LLMs, indicating that the plug-in model's predictionis the primary signal for ReasonTSC. Howerver, the effect of components including confidence and dataset level accuracy can not be ommit, as they play axuliary information for LLM decision. Removing logits leads to consistent performance drops for all three LLMs in Figure 8 . Furthermore, Figure 9 analyzes the reasons behind LLM overrides of plug-in predictions. GPT, Llama, and DeepSeek consider confidence scores in 36.51%, 50.00%, and 31.91% of override cases, respectively. DeepSeek and Llama additionally factor in dataset-level accuracy when choosing to override the plug-in model. These results suggest that while the plug-in prediction is the dominant signal, confidence and dataset-level accuracy contribute meaningfully rather than causing overfitting, and together help the LLM make more reliable corrections.

---

> > ### Author Response · Authors · 2025-11-21
> >
> > > **W3&Q2: Concerns for baseline comparision**
> >
> > In the main experiment, we compare against vanilla CoT as the ICLR'25 paper 'Can LLMs Understand Time Series Anomalies?' shows that explicit reasoning generally fails to improve anomaly detection, and our results with self-consistency and self-correction (Fig. 2) further confirm that naively applying generic reasoning techniques does not yield satisfactory performance. We demonstrate that a properly designed reasoning framework, LLMs can indeed benefit from structured reasoning for time series understanding.
> >
> > We appreciate the suggestion to include stronger reasoning and tool-use baselines such as tree-based search, program-of-thought, retrieval-augmented prompts, or statistical/spectral tool calls. However, these methods typically introduce substantial additional budgets (e.g., multiple reasoning branches, retrieval operations, or heavy tool invocation), making fair comparison under matched tokens and latency nontrivial. The cost gap between ReasonTSC and vanilla CoT is modest (please see Appendix B.8), and ReasonTSC with a plug-in classifier can identify and correct the classifier’s errors, offering a cost-effective approach while maintaining strong performance.
> >
> > ||Time(s)|Time(s)|Cost|Cost|
> > |----------|---------|--------|--------|--------|
> > |Dataset|ReasonTSC|Vanilla CoT|ReasonTSC|Vanilla CoT|
> > |Dist.TW|39.53|25.35|0.029|0.017|
> > |Med.Img|35.05|49.65|0.043|0.029|
> > |BME|36.79|36.36|0.027|0.017|
> > |Arr.Hd|79.34|40.15|0.040|0.021|
> > |Dod.LD|30.63|33.28|0.044|0.026|
> > |Elec.|30.05|28.61|0.035|0.019|
> >
> > > **W4: Concerns for effectiveness across LLMs**
> >
> > In the main experiment, we select three representative LLMs covering different parameter scales for evaluation. To further support the breadth of effectiveness, we additionally evaluate six mainstream LLMs across three datasets under the same experimental setting, with the complete results are provided in Table 9 (please refer to Appendix B.7), demonstrating the robustness of the ReasonTSC framework. We appreciate the suggestion and will include more comprehensive measurements of cost, latency, and context length in the revised manuscript to ensure clearer and more comparable reporting.
> >
> > > **W5: Concerns for override behaviors**
> >
> > Thank you for raising this concern. Prior work has shown that integrating LLMs with smaller discriminative models is an effective design choice that reduces hallucinations and stabilizes in-context learning [1,2]. In our setting, however, the overturn behavior cannot be attributed to simple confidence reweighting. As shown in Table 3, different LLMs exhibit distinct override tendencies, indicating model-specific reasoning rather than a uniform reaction to injected numbers. Our analysis in Figure 9 further shows that GPT, Llama, and DeepSeek override the plug-in in 63.49%, 48.05%, and 51.97% of cases because the plug-in's answer conflicts with their own understanding of time series patterns, indicating that the plug-in's confidence contributes only partially to the model’s decision-making. The ablation in Figure 8(b) also demonstrates that removing plug-in logits increases the override rate, implying that the plug-in does not dominate the decision process. Instead, LLMs rely more heavily on their own interpretation of the time series when fewer auxiliary signals are available.
> >
> > [1] Supervised Knowledge Makes Large Language Models Better In-context Learners (ICLR24)
> >
> > [2] Small Models are Valuable Plug-ins for Large Language Models (ACL24 Findings)
> >
> >
> > > **W6&Q3: Concerns for framework generalization**
> >
> > We appreciate your suggestions and will revise our conclusions and related statements accordingly. In the main paper, ReasonTSC focuses on univariate time series and evaluates 15 UCR/UEA datasets spanning diverse domains. These datasets cover between 3 and 15 classes and include 80 to 288 samples per class. Under the 2-shot setting, the total input length ranges from 160 to 4032, corresponding to approximately 1.8k to 12k tokens (please see Table 11 in Appendix D.11). We believe the breadth of datasets and the depth of evaluation provide solid empirical evidence for our claims.
> >
> > Regarding generalization, we agree that the multivariate setting differs from the univariate case due to cross-variable dependencies and longer temporal horizons. Reasoning in this context is therefore not a trivial extension and warrants further investigation. We present a solution for adapting ReasonTSC to multivariate time series in Appendix B.7. In the multivariate setting, On Epilp. dataset (4 classes, 3 dimensions, 206 time points per dimension), ReasonTSC achieves 80.70% (DeepSeek-R1) and 73.68% (GPT-4o-mini), both exceeding the plug-in model MOMENT (71.93%). In addition, ReasonTSC generalizes beyond classification and can be applied to other time series tasks, including anomaly detection and vision-based reasoning (Please see detailed results in Appendix B.6).

---

### Official Review · Reviewer_AF2y · 2025-10-30

**Soundness:** 2
**Presentation:** 3
**Contribution:** 2
**Rating:** 2
**Confidence:** 3

**Summary:**

This paper proposes ReasonTSC, a novel framework that enhances Large Language Model (LLM) reasoning for Time Series Classification (TSC) tasks. The key idea is to tailor multi-turn reasoning to the characteristics of time-series data (trend, cyclicity, amplitude, etc.) and fuse decisions with outputs from pretrained time-series foundation models (TSFMs) such as MOMENT and Chronos.

**Strengths:**

pros:
1. The authors provide a convincing argument for why directly transferring NLP-style reasoning to time-series tasks underperforms. The identification of two research questions (steering reasoning and fusing external knowledge) is logically grounded.

2. The evaluation covers 15 UCR/UEA datasets, multiple LLMs, and two foundation models. good job for comprehensive experiments; The experimental design is broad, and the authors include ablations and analysis (Figures 7–9, Tables 1–3) that explore how reasoning rounds, pattern prompts, and fusion impact performance.

3. Readable and structured presentation. The writing is clear and the reasoning process (Figure 1) is well visualized. The inclusion of illustrative rationales (Appendix C.2) makes the framework interpretable.

4. Empirical novelty. The “fused decision” idea of combining plug-in model confidence scores with reasoning traces is practically useful and may inspire further LLM–TSFM hybrid methods.

**Weaknesses:**

Cons:
1. Lack of methodological depth; The framework is mostly prompt-engineering-based, without introducing a new model or learning mechanism. The multi-turn reasoning steps (pattern analysis → plug-in fusion → backtracking) are intuitive but not theoretically justified or derived from a formal reasoning principle.

2. Evaluation setup may inflate improvements. Baselines like TimesNet and FEDformer are fully trained, while ReasonTSC uses few-shot inference, making direct percentage comparisons misleading.

3. The “90% improvement” figure comes from very low-performing vanilla CoT baselines, so relative gains are exaggerated.


4. All evaluations are on relatively small-scale UCR/UEA datasets. There is no demonstration on long or noisy industrial time series (e.g., finance, sensor, traffic).

5. The approach depends heavily on token budgets and handcrafted prompts, which limits scalability beyond short sequences (<10 k tokens).

6. Ablation insufficient for causal insight
- Although Figures 12–15 show performance drops when removing reasoning turns, there is no clear causal analysis of why these stages help (e.g., how much each turn contributes semantically).
- The distinction between pattern understanding and fusion reasoning remains qualitative.

7. Limited novelty relative to existing reasoning works; Compared to frameworks like Self-Refine, Program-Aided Reasoning, or TS-Reasoner, ReasonTSC mainly reorganizes prompting stages rather than proposing a new reasoning paradigm. It lacks a deeper exploration of reasoning consistency, explainability metrics, or learning-based refinement that would strengthen its contribution.

**Questions:**

How much of the improvement comes from the reasoning process itself versus simply using the plug-in model’s logits and confidence scores?

Could you include a fairer baseline where the plug-in model outputs are given to the LLM without multi-turn reasoning?

How scalable is ReasonTSC to long or multivariate time series (e.g., >10 k tokens or >10 variables)?

Have you tested ReasonTSC on any real-world datasets beyond UCR/UEA, such as financial or healthcare data?

How sensitive are the results to prompt wording or model choice?

Can you quantify reasoning quality—e.g., does more reasoning always lead to better accuracy?

Could you show an ablation for each reasoning round separately (1st only, 1 + 2, 1 + 3, etc.)?

How exactly are the plug-in logits fused with the LLM output—is it a weighted average, rule-based, or purely textual reasoning?

Are the identified “time-series patterns” verified to be meaningful or correlated with true features?

How does this approach differ conceptually from existing iterative reasoning methods such as Self-Refine or Program-Aided Reasoning?

---

> ### Author Response · Authors · 2025-11-21
>
> > **W1&W7&Q10: Concerns for novelty and baselines**
>
> Thank you for raising this concern. Our work does not claim to introduce a new model architecture or learning mechanism. Instead, the contribution focuses on addressing a domain-level gap that existing reasoning techniques have not effectively resolved. Time series (TS) is a structure-rich, non-text modality whose pattern semantics differ fundamentally from natural language. This creates an open question that prior works have not addressed: why NLP-oriented reasoning strategies transfer poorly to TS tasks, and how TS-specific reasoning can be reliably elicited without additional training.
>
> Unlike symbolic or mathematical reasoning, TS classification lacks a formal logic that LLMs can easily exploit. Progress therefore requires identifying modality-specific failure modes and developing elicitation strategies that activate the necessary reasoning behaviors. ReasonTSC moves in this direction by explicitly targeting three such modes:
>
> (1) LLMs often fail to adopt pattern-oriented thinking, while pattern primitives are essential in TS. Turn 1 explicitly steers the model toward pattern-level analysis.
>
> (2) LLMs exhibit hallucinated certainty and rarely reconsider alternatives. Turn 3 introduces a backtracking mechanism to mitigate this issue.
>
> (3) Naïve few-shot prompting does not directly improve performance due to modality mismatch. Our fused decision strategy integrates TSFM outputs to provide a more reliable signal.
>
> To our knowledge, ReasonTSC is the first framework showing that LLMs can successfully correct TSFM prediction errors through TS-specific reasoning. This suggests that existing prompting or CoT variants do not trigger. The generality of this insight is supported by our large-scale evaluation across 16 LLMs, 15 datasets, and two TS foundation models.
>
> Regarding baselines, we have included self-refine[6] (denoted as self-correction) by setting three reasoning rounds (initial answer plus two refinement steps) in Figure 2. For Program-Aided Reasoning, its core idea is to levarage program aided language models to provide code-based prompts. This paradigm is suitable for symbolic or algorithmic tasks. However, time series cannot be meaningfully represented or directly interprated in code-based form. For TS-Reasoner, the work was posted on arXiv in October and, to the best of our knowledge, does not provide open-source code or released model checkpoints. we believe it is not feasible to include it as a baseline at this stage.
>
> Finally, we would like to remark that there are several concurrent submissions to ICLR 2026 [1–5], which collectively indicate that time series reasoning is an emerging and timely research direction. These works explore valuable and complementary perspectives, such as RL-based reasoning enhancement, modality alignment through encoders, and incorporating the vision modality to enrich TS semantics. Our work investigates whether TS-specific reasoning can be reliably elicited through prompting alone, without additional LLM training or resource-intensive model adaptation. Therefore, our approach serves as a distinct and orthogonal contribution that focuses on activating the TS-reasoning capabilities inherent in LLMs, and the presence of these concurrent ICLR submissions does not diminish the significance of our work.
>
> [1] TimeOmni-1: Incentivizing Complex Reasoning with Time Series in Large Language Models
>
> [2] Unlocking the Value of Text: Event-Driven Reasoning and Multi-Level Alignment for Time Series Forecasting
>
> [3] From Images to Signals: Are Large Vision Models Useful for Time Series Analysis?
>
> [4] AnomSeer: Reinforcing Multimodal LLMs to Reason for Time-Series Anomaly Detection
>
> [5] Time Series Forecasting as Reasoning: A Slow-Thinking Approach with Reinforced LLMs
>
> [6] Self-refine: Iterative refinement with self-feedback. In Thirty-seventh Conference on Neural Information Processing Systems, 2023.
>
> > **W2: Concerns for evaluation setup**
>
> In Table 4 (Appendix B.1), baselines such as TimesNet and FEDformer are fully trained on the entire dataset, whereas ReasonTSC operates in a few-shot inference setting. Our intention is not to claim a direct percentage-to-percentage superiority over these task-specific models, but to show that with only a few in-context demonstrations, ReasonTSC can outperform its own plug-in classifier, correct its errors, and achieve performance comparable to fully trained baselines on most subsets. We have modified this in the revised manuscript (lines 907–909) to prevent potential misinterpretation.

---

> > ### Author Response · Authors · 2025-11-21
> >
> > > **W3: Concerns for performance gains**
> >
> > Thank you for raising this point. We agree that 'vanilla CoT' is a weak baseline, and this is precisely why we examine it: the ICLR'25 paper 'Can LLMs Understand Time Series Anomalies?' concludes that explicit reasoning fails to improve LLMs' time series anomaly detection capabilities. Our results show that this conclusion stems from the limitations of the vanilla CoT setup itself. With a properly designed reasoning framework, LLMs can indeed benefit from structured reasoning for time series understanding. Besides, we compare other reasoning strategies (Self-Consistency, Self-Correction) in figure 2. These comparisons indicate that naively applying generic reasoning techniques does not yield satisfactory performance, underscoring the need for a tailored framework. To provide a more complete and fair comparison, we have added the improvement over the plug-in models in Table 1 and Table 2 in the revised manuscript.
> >
> > | Model | Dist.TW | Mid.TW | Mid.OA | Elec.| Med.Img | BME | Arr.Hd | Dod.LD  |
> > |----------|---------|--------|--------|------|----------|-----|--------|---------|
> > | MOMENT | 62.59 | 51.30 | 60.39 | 57.89 | 76.97 | 74.00 | 65.71 | 31.17 |
> > | ReasonTSC | 65.71 | 57.42 | 63.64 | 67.11 | 80.26 | 82.67 | 69.14 | 38.96 |
> > | Improvement vs. MOMENT | +4.98% | +11.93% | +5.38% | +15.93% | +4.27% | +11.72% | +5.22% | +24.99%  |
> > |  **Model** | **CBF** | **Rkt.Spt** | **ERing** | **Nt.Ops** | **Lbr.** | **Eplp.** | **Pen.** | **Avg** |
> > | MOMENT | 66.00 | 59.21 | 72.59 | 65.56 | 48.49 | 88.40 | 85.62 | 64.39 |
> > | ReasonTSC | 74.00 | 63.16 | 74.07 | 67.78 | 55.00 | 91.30 | 86.30 | 69.10 |
> > | Improvement vs. MOMENT | +12.12% | +6.67% | +2.04% | +3.39% | +13.43% | +3.28% | +0.79% | +7.31%  |
> >
> >
> > > **W4&Q4: Concerns for dataset**
> >
> > The UCR/UEA Archive is a widely used benchmark for time-series classification, and our selected 13 real-world and 2 synthetic datasets cover diverse domains such as medical, energy, traffic, and gesture (please refer to Table 11 in Appendix D.1). These datasets also include varying sequence lengths (per category), ranging from 80 to 288 time points. This results in total input lengths (number of categories × length) between 160 and 4032 time points under the 2-shot setting.
> >
> > > **W5&Q3: Concerns for long or multivariate settings**
> >
> > Thank you for the thoughtful comment. Across the selected UCR/UEA datasets, the token counts of ReasonTSC range from 1.8k to 12k. As shown in Figure 16 (please refer to Appendix B.5), its performance remains stable across different numbers of categories, sequence lengths, and token budgets, consistently yielding improvements. We discuss scalability explicitly and provide a practical extension for multivariate time series in Appendix B.6.
> >
> > ```
> > - We use <dim> tags to separate different dimensions within each sample;
> >
> > - We employ interval sampling on the time series data to mitigate extremely long sequences by concatenating all dimensions.;
> >
> > - We adapt the prompt to guide the LLM in comparing dimensions separately across categories in the first round of pattern reasoning.
> > ```
> > On Epilp. dataset (4 classes, 3 dimensions, 206 time points per dimension), using 1:4 interval sampling, ReasonTSC achieves 80.70% (DeepSeek-R1) and 73.68% (GPT-4o-mini), both exceeding the plug-in model MOMENT (71.93%). This suggests that ReasonTSC remains effective when reasoning over multivariate or long time series.
> >
> > > **W6: Cocerns for ablation results**
> >
> > We appreciate your concerns. We would like to clarify that Figures 12–15 in B.3 and B.4 serve as supplementary ablations to the main results (Figures 4–8) and are designed to isolate the causal roles of each reasoning stage in ReasonTSC. ReasonTSC contains three rounds: TS pattern reasoning, plug-in fusion reasoning, and integrative step-by-step reasoning. The ablations directly evaluate their semantic contribution. In Figures 12 and 13, removing the second or third round causes clear performance drops, showing that plug-in fusion provides useful behavioral cues that help the LLM correct the plug-in's systematic errors, while the integrative CoT stage enables self-refinement and mitigates the model's own incorrect initial predictions. Figures 14 and 15 further compare ReasonTSC's tailored CoT with vanilla CoT, demonstrating that integrative step-by-step reasoning consistently yields larger gains over plain LLMs. Notably, removing logits or the entire plug-in yields improvement ratios more than twice those of vanilla CoT (55.63% without logits and 51.45% without the full plug-in), which highlights the stronger causal effect of ReasonTSC's customized reasoning strategy. We will refine the presentation in the revision to make these causal conclusions more explicit.

---

> > > ### Author Response · Authors · 2025-11-21
> > >
> > > > **Q1&2: Concerns for plug-in model**
> > >
> > > Figure 8(a) illustrates the performance gain brought by the plug-in model within ReasonTSC. To further clarify this effect, Figure 15 (please refer to Appendix B.3) compares ReasonTSC's CoT reasoning with vanilla CoT under settings with and without the plug-in model. Using Llama as an example, the results show that even without multi-turn reasoning, ReasonTSC's CoT surpasses vanilla CoT. Incorporating plug-in model information yields an additional improvement, but it still does not match the full ReasonTSC performance. These findings indicate that while the plug-in model contributes substantially, the benefits from ReasonTSC's pattern-aware and integrative step-by-step reasoning remain essential and cannot be discounted.
> > >
> > >
> > > |||Dist.TW |Mid.OA|Dod.LD|Med.Img|
> > > |----------|---------|--------|--------|------|------|
> > > | Vanilla CoT|w/o plug-in|26.62|18.83|23.38|11.84|
> > > ||with plug-in|62.59| 50.55|57.14|69.08|
> > > |ReasonTSC's CoT|w/o plug-in|38.13|27.92|36.36|18.42|
> > > ||with plug-in|64.03|53.90|57.14|71.05|
> > >
> > >
> > > > **Q5: Concerns for performanmce robustness**
> > >
> > > Thank you for raising this concern. To assess prompt sensitivity, we design an additional experiment in which DeepSeek-R1 is asked to paraphrase ReasonTSC’s original prompt while preserving its semantic intent (please see Appendix B.5). Using this paraphrased prompt, ReasonTSC with GPT-4o-mini produces consistent predictions across both prompt variants, indicating strong robustness to prompt perturbations. For model selection, we integrate two representative TSFMs (MOMENT and Chronos) as plug-in classifiers. As shown in Table 1 and Table 2, ReasonTSC maintains stable performance across both models.
> > >
> > > | Dataset | Original | Paraphrased | Prediction Changes |
> > > |----------|---------|--------|--------|
> > > | Dist.TW | 63.31 | 62.59 | 1 |
> > > | Mid.TW | 57.79 | 57.79 | 0 |
> > > | BME | 77.33 | 80.67 | 3 |
> > > | Dod.LD | 31.17 | 32.47 | 2 |
> > >
> > >
> > > > **Q6: Concerns for reasoning quality**
> > >
> > > In our results, it is observed that DeepSeek presents relatively better performance under the ReasonTSC framework compared to Llama and GPT. This might stem from DeepSeek's significantly larger parameter size (671B for DeepSeek-R1 vs. 8B for GPT-4o-mini and 70B for Llama) as well as its specialized reasoning-enhanced training. Thus the reasoning quality is influenced by factors such as LLM parameters, release dates, and whether reasoning-focused post-training techniques were applied.
> > >
> > >
> > > > **Q7: Concerns for each reasoning round**
> > >
> > > Thank for your concern. Figures 12 and 13 (please refer to Appendix B.3) provide ablations isolating the contribution of each reasoning round. Specifically, we remove (i) the plug-in model behavior analysis (1+3) and (ii) the integrative step-by-step reasoning (1+2). The results indicates that the combined effect of the three reasoning stages can surpass the performance of single stage. In addition, the notable drop in override accuracy in these ablations validates that ReasonTSC relies on the combination of pattern reasoning and plug-in model analysis to reliably correct TSFM prediction errors.
> > >
> > >
> > > > **Q8: Concerns for the fusion method**
> > >
> > > For each dataset, we first train the TSFM on the training split and obtain its predictions, logits, and accuracy on the test split. For every test sample, these plug-in information (predicted label, per-class logits, and model accuracy) are inserted into the prompt. The LLM then reasons over this information through text. Please see the full prompt template and fusion format in Appendix C.1.
> > >
> > >
> > > > **Q9: Concerns for time series patterns**
> > >
> > > Thank for your concern. Our pattern selection strategy is grounded in existing time series literature [1][2][3], which has proven the fundamental role and effectiveness of these patterns. Empirically, Figure 7 shows that removing the top-ranked patterns leads to a clear performance drop for ReasonTSC with DeepSeek-R1, while removing random patterns results in performance comparable to the full model. This contrast indicates that the identified patterns are indeed meaningful and aligned with true discriminative features.
> > >
> > > [1] ChatTS: Understanding, Chat, Reasoning about Time Series with TS-MLLM (VLDB'25)
> > >
> > > [2] Language Models Still Struggle to Zero-shot Reason about Time Series (EMNLP'24)
> > >
> > > [3] Timeseriesexam: A time series understanding exam[J].

---

### Official Review · Reviewer_9a93 · 2025-11-01

**Soundness:** 3
**Presentation:** 3
**Contribution:** 1
**Rating:** 4
**Confidence:** 5

**Summary:**

The paper identifies that standard Large Language Model (LLM) reasoning techniques, like simple Chain-of-Thought (CoT), fail to improve performance on time series classification (TSC) tasks. The authors propose ReasonTSC, a novel framework to elicit more effective, domain-specific reasoning from LLMs without retraining them.

**Strengths:**

**Excellent Empirical Results:** The primary strength of the paper is its impressive results. Tables 1 and 2 show that ReasonTSC achieves massive performance gains (e.g., +87% to +686% on some datasets) compared to the "Vanilla CoT" baseline.

**Novel Fusion Strategy:** The core idea of "fusing" the logits and predictions of a specialized model into the LLM's reasoning context is a clever and practical way to inject deep domain knowledge without requiring any fine-tuning of the LLM.

**Strong Ablation Studies:** The authors do a commendable job of demonstrating why their framework works. They provide extensive analyses showing:

- That ReasonTSC can successfully identify and correct the plug-in model's false predictions (Table 3).

- That the "backtracking" step of considering alternatives is crucial (Fig 5).

- That the "fused decision" (using the plug-in) is critical to performance (Fig 8).

**Weaknesses:**

**Lack of Fundamental Novelty:** The paper's novelty is questionable. The framework is an engineering contribution, not a fundamental one. It skillfully combines several existing techniques:

- Few-shot In-Context Learning (Turn 1)

- Tool-Assisted Reasoning (Turn 2, where the plug-in model is the "tool")

- Multi-step Chain-of-Thought with backtracking/self-refinement (Turn 3)

The "tailored thinking" is essentially just a very specific, well-written set of prompts. While effective, it doesn't introduce a new reasoning mechanism but rather a new, complex application of existing ones.



**Unverifiable Claims and Over-Attribution:** This is the most significant weakness, which you correctly identified. The paper claims the LLM "evaluates the plug-in model’s confidence" and "adjusts [its] understanding."

- This claim is unfalsifiable. We have no way to verify that the LLM is actually "evaluating confidence" in a human-like sense.

- The LLM is a black-box text generator. It is far more likely that it is not performing this complex meta-reasoning, but is simply using the logits and prediction as powerful new features or "hints" in its context.

- The paper consistently attributes human-like cognition to the model (e.g., "it recognizes," "it tends to present... interpretations") when it is simply generating text that mimics those cognitive processes. The "rationales" it produces (e.g., in Sec C.2) are just plausible-sounding text, not a verifiable trace of its internal reasoning.

**Weak Baseline:** The massive performance gains are made possible by comparing against a "Vanilla CoT" (just adding "please think step by step"). This is a very weak, zero-shot, no-context baseline. It's not surprising that a complex, multi-turn, few-shot framework that also uses a specialized, pre-trained model for hints (ReasonTSC) would win.

**High Complexity and Cost:** The proposed framework is extremely complex and resource-intensive. It requires:

 - A separate, trained plug-in model (MOMENT/Chronos) to be run first.

- Multiple, very long prompts (with 2-shot and 3-shot examples, plus logits) to be fed to the LLM.

- Multiple "turns" or API calls to the LLM.

This makes it far more expensive than a simple classification model.

**Questions:**

See weakness

---

> ### Author Response · Authors · 2025-11-21
>
> > **W1: Novelty concerns**
>
> Thank you for raising this concern. Our work does not propose a new formal reasoning paradigm. The contribution instead lies in addressing a domain-level gap that existing reasoning techniques have not effectively resolved. Time series (TS) is a structure-rich, non-text modality whose pattern semantics differ fundamentally from natural language. This leads to a practical and scientific question that prior methods have not answered: why NLP-oriented reasoning strategies underperform on TS tasks, and how LLMs' TS-specific reasoning can be reliably elicited without additional training.
>
> Unlike symbolic or mathematical domains, TS classification lacks a well-established formal logic. Progress therefore depends on identifying the failure modes of LLMs in this modality and designing elicitation strategies that activate the necessary reasoning behaviors. Our work makes a step in this direction by introducing a framework that systematically elicits TS-specific reasoning through:
>
> (1) LLMs often fail to adopt pattern-oriented thinking, while pattern primitives are essential in TS. Turn 1 explicitly steers the model toward pattern-level analysis.
>
> (2) LLMs exhibit hallucinated certainty and rarely reconsider alternatives. Turn 3 introduces a backtracking mechanism to mitigate this issue.
>
> (3) Naïve few-shot prompting does not directly improve performance due to modality mismatch. Our fused decision strategy integrates TSFM outputs to provide a more reliable signal.
>
> ReasonTSC is, to our knowledge, the first framework showing that LLMs can successfully correct TSFM prediction errors. This suggests that the proposed elicitation structure activates latent TS-specific reasoning abilities that existing prompting or CoT variants do not trigger. Finally, the generality of the approach is supported by the scale of our evaluation: 16 LLMs, 15 datasets, and two types of TS foundation models. This indicates that the framework is not a narrow engineering combination of known techniques, but a transferable strategy for reasoning over TS modalities.
>
> Finally, we would like to remark that there are several concurrent submissions to ICLR 2026 [1–4], which collectively indicate that time series reasoning is an emerging and timely research direction. These works explore valuable and complementary perspectives, such as RL-based reasoning enhancement, modality alignment through encoders, and incorporating the vision modality to enrich TS semantics. Our work investigates whether TS-specific reasoning can be reliably elicited through prompting alone, without additional LLM training or resource-intensive model adaptation. Therefore, our approach serves as a distinct and orthogonal contribution that focuses on activating the TS-reasoning capabilities inherent in LLMs, and the presence of these concurrent ICLR submissions does not diminish the significance of our work.
>
> [1] TimeOmni-1: Incentivizing Complex Reasoning with Time Series in Large Language Models
>
> [2] Unlocking the Value of Text: Event-Driven Reasoning and Multi-Level Alignment for Time Series Forecasting
>
> [3] From Images to Signals: Are Large Vision Models Useful for Time Series Analysis?
>
> [4] AnomSeer: Reinforcing Multimodal LLMs to Reason for Time-Series Anomaly Detection
>
> > **W2: Concerns for override behavior interpretation**
>
> Thank you for raising this concern. We agree that LLMs are black-box text generators and we do not claim access to their internal reasoning or human-like cognitive processes. Our analysis focuses strictly on observable behavioral signals. Prior work has shown that incorporating smaller discriminative models can provide useful auxiliary cues for LLMs [1,2]. In our case, the override behavior cannot be explained solely by features or 'hints'. Table 3 shows that different LLMs exhibit distinct override tendencies, suggesting heterogeneous decision behaviors rather than a uniform response to injected logits. Our analysis in Figure 9 shows that GPT, Llama, and DeepSeek override the plug-in in 63.49%, 48.05%, and 51.97% of cases because the plug-in's answer conflicts with their own understanding of time series patterns, indicating that the plug-in's confidence contributes only partially to the model’s decision-making. The ablation in Figure 8(b) shows that removing plug-in logits increases the override rate, implying that the plug-in does not dominate the decision process. Regarding the rationales in Sec. C.2, we do not interpret them as literal explanations of internal reasoning, rather, they serve as a qualitative lens to examine whether the generated justifications align with recognizable time series patterns. We will follow your suggestion and revise our phrasing to describe model behavior more cautiously.
>
> [1] Supervised Knowledge Makes Large Language Models Better In-context Learners (ICLR24)
>
> [2] Small Models are Valuable Plug-ins for Large Language Models (ACL24 Findings)

---

> > ### Author Response · Authors · 2025-11-21
> >
> > > **W3: Concerns for performance gains**
> >
> > Thank you for the thoughtful comment. We agree that 'vanilla CoT' is a weak baseline, and this is precisely why we include it: the ICLR'25 paper 'Can LLMs Understand Time Series Anomalies?' claims that explicit reasoning fails to improve LLMs' time series anomaly detection capabilities. Our results show that this conclusion stems from the limitations of the vanilla CoT setup itself. With a properly designed reasoning framework, LLMs can indeed benefit from structured reasoning for time series understanding. Besides, we compare other reasoning strategies (Self-Consistency, Self-Correction) in figure 2. These comparisons indicate that naively applying generic reasoning techniques does not yield satisfactory performance, underscoring the need for a tailored framework. Our ablation studies demonstrate that each component of ReasonTSC contributes to the overall improvement. Moreover, ReasonTSC consistently outperforms the plug-in models used for hint generation, demonstrating that the framework can correct their errors rather than merely inheriting their predictions. To provide a more complete and fair comparison, we have added the improvement over the plug-in models in Table 1 and Table 2 in the revised manuscript.
> >
> >
> > | Model | Dist.TW | Mid.TW | Mid.OA | Elec.| Med.Img | BME | Arr.Hd | Dod.LD  |
> > |----------|---------|--------|--------|------|----------|-----|--------|---------|
> > | MOMENT | 62.59 | 51.30 | 60.39 | 57.89 | 76.97 | 74.00 | 65.71 | 31.17 |
> > | ReasonTSC | 65.71 | 57.42 | 63.64 | 67.11 | 80.26 | 82.67 | 69.14 | 38.96 |
> > | Improvement vs. MOMENT | +4.98% | +11.93% | +5.38% | +15.93% | +4.27% | +11.72% | +5.22% | +24.99%  |
> > |  **Model** | **CBF** | **Rkt.Spt** | **ERing** | **Nt.Ops** | **Lbr.** | **Eplp.** | **Pen.** | **Avg** |
> > | MOMENT | 66.00 | 59.21 | 72.59 | 65.56 | 48.49 | 88.40 | 85.62 | 64.39 |
> > | ReasonTSC | 74.00 | 63.16 | 74.07 | 67.78 | 55.00 | 91.30 | 86.30 | 69.10 |
> > | Improvement vs. MOMENT | +12.12% | +6.67% | +2.04% | +3.39% | +13.43% | +3.28% | +0.79% | +7.31%  |
> >
> >
> > > **W4: Concerns for cost**
> >
> > Thank you for raising this concern. To quantify cost, we measure the per-sample inference time and API cost of ReasonTSC and Vanilla CoT (please see Appendix B.8). The overall overhead remains comparable because the prompt length is dominated by the time-series input rather than the number of reasoning turns. ReasonTSC achieves substantially higher accuracy while keeping the additional cost tolerable a practical range.
> >
> > ||Time(s)|Time(s)|Cost|Cost|
> > |----------|---------|--------|--------|--------|
> > |Dataset|ReasonTSC|Vanilla CoT|ReasonTSC|Vanilla CoT|
> > |Dist.TW|39.53|25.35|0.029|0.017|
> > |Med.Img|35.05|49.65|0.043|0.029|
> > |BME|36.79|36.36|0.027|0.017|
> > |Arr.Hd|79.34|40.15|0.040|0.021|
> > |Dod.LD|30.63|33.28|0.044|0.026|
> > |Elec.|30.05|28.61|0.035|0.019|
> >
> > Regarding the plug-in TSFMs, their use does not substantially increase system complexity: models such as MOMENT (0.8B) and Chronos (710M) are pretrained foundation models with cross-task generalization, and their task-specific fine-tuning and inference cost is relatively small compared with training a TSFM from scratch. More broadly, leveraging LLMs for TS reasoning has emerged as a promising research direction, as LLMs can provide competitive zero-/few-shot performance and interpretable reasoning traces that traditional supervised models cannot offer. Recent work has highlighted this trend across both prompting-based and model-based approaches [1][2][3], suggesting that exploring LLM-centric methods is timely and well-motivated.
> >
> > [1] Large language models are zero-shot time series forecasters. Advances in Neural Information Processing Systems, 2023.
> >
> > [2] ChatTS: Aligning Time Series with LLMs via
> > Synthetic Data for Enhanced Understanding and Reasoning. In Proceedings of
> > the VLDB Endowment, 2025.
> >
> > [3] Lstprompt: Large language models as zero-shot time
> > series forecasters by long-short-term prompting. Findings of the Association for Computational Linguistics: ACL 2024.

---

### Author Response · Authors · 2025-12-03

Dear Area Chairs, Senior Area Chairs, and Program Chairs,

Thank you for overseeing the review process of our submission. We sincerely appreciate the reviewers' constructive feedback. Below we provide a concise summary of the key points addressed in our rebuttal.

## Paper Overview and Strengths

Our paper introduces ReasonTSC, a novel framework for leveraging LLMs in time series classification through tailored multi-turn reasoning and a fused decision-making strategy. ReasonTSC consistently outperforms standard prompting baselines and is able to correct errors of plug-in models. The results demonstrates that, with appropriate guidance, LLMs can reason effectively over time series data without requiring visual encoders or retraining.

Reviewers highlighted several strengths:

* **Effective design**: The core idea of 'fusing' the logits and predictions of a specialized model into the LLM's reasoning context is a clever and practical way to inject deep domain knowledge without requiring any fine-tuning of the LLM. (Reviewers 9a93, AF2y, Q2KJ)

* **Strong empirical results**: ReasonTSC achieves excellent and consistent improvements covers 15 UCR/UEA datasets, multiple LLMs, and two foundation models. (Reviewers 9a93, AF2y)

* **Comprehensive ablations**: Ablations on override behavior, backtracking, and fusion components effectively demonstrate why the framework works.  (Reviewers 9a93, JuQM)

* **Readable and structured presentation**: Readable and structured presentation. The writing is clear and the reasoning process is well visualized. The inclusion of illustrative rationales makes the framework interpretable. (Reviewers AF2y, JuQM)


## Summary of Reviewer Concerns and Our Responses

### 1. Concerns for Novelty and Methodological Depth

Reviewers 9a93, AF2y, and Q2KJ concerns that ReasonTSC is engineering combination rather than proposing a new reasoning paradigm.

**Response:** Our contribution lies in addressing a domain-specific gap: NLP-oriented reasoning performs poorly on time series data due to modality mismatch. Unlike symbolic or mathematical domains, TS classification lacks a well-established formal logic. We identify three specific failure modes (lack of pattern-oriented analysis, hallucinated certainty, and weak few-shot transfer) and design a strategy that reliably elicits TS-specific reasoning without additional LLM training or resource-intensive model adaptation. Our results across 16 LLMs and 2 TSFMs show that the proposed framework activates TS-specific reasoning that existing prompting or CoT variants do not trigger, and can successfully correct TSFM prediction errors. Besides, recent concurrent work further confirms that TS reasoning is an emerging area while our method provides a distinct and orthogonal contribution.

### 2. Concerns for Cost and Complexity

Reviewers 9a93, AF2y and JuQM raise concerns that the multi-turn design may be complex or resource-intensive.

**Response:** We measure per-sample inference time and API cost for both ReasonTSC and vanilla CoT. The token budget is dominated by time-series tokens, while text tokens (instructions and reasoning) form only a small fraction. In the first round, both methods use the same number of TS examples (two per class), while the text-token portion remains relatively small and comparable.Thus, ReasonTSC achieves better performance without introducing a substantial increase in the proportion of text tokens, keeping the additional amount of tokens within an acceptable range relative to Vanilla CoT.

### 3. Concerns for Baseline Comparisons and Performance Gains

Reviewers 9a93 and AF2y raise concerns about the baseline selection and whether the performance gains come from the choice of baselines.

**Response:** We include Vanilla CoT because prior work claims that explicit reasoning does not help time series tasks, and our goal is to show that LLMs can benefit from structured reasoning with an appropriately designed framework. We also compare other reasoning strategies such as Self-Consistency and Self-Correction. In addition, ReasonTSC consistently outperforms the plug-in models, demonstrating that the framework can correct their errors rather than merely inheriting their predictions. To ensure fairness, we have added the improvements over the plug-in models in Table 1 and Table 2 of the revised manuscript.

---

> ### Author Response · Authors · 2025-12-03
>
> ### 4. Concerns for Framework Generalization
>
> Reveiwers AF2y, Q2KJ and JuQM raise concerns about ReasonTSC's generalization to multivariate time series, long-horizon settings, and other time series tasks.
>
> **Response:** ReaonTSC's performance remains stable across varying sequence lengths and token budgets. We explicitly discuss scalability and show that ReasonTSC can be easily extended to multivariate settings via dimension tagging and interval sampling, achieving improved performance. In addition, ReasonTSC can be applied to other time series tasks, including anomaly detection and vision-based reasoning (see Appendix B.6). These results indicate that ReasonTSC effectively generalizes to multivariate time series and diverse tasks.
>
> ### 5. Concerns for Interpretation of LLM Override Behavior
>
> Reviewer Q2KJ and AF2y raise concerns that the observed human-like reasoning might mainly stem from plug-in hints, with the language model reacting to confidence patterns already visible in the injected numbers.
>
> **Response:** Our experiment results show that different LLMs exhibit distinct override tendencies, indicating model-specific reasoning rather than a uniform reaction to injected numbers. Figure 8(b) further demonstrates that removing plug-in logits increases the override rate, suggesting that the plug-in model does not dominate the decision process, indicating that LLMs rely more on their own interpretation of the time series when fewer auxiliary signals are available.
>
> We are sorry to hear about the recent OpenReview bug issue and fully support the proposed remedial actions. Due to the updated policy of ICLR-26, we understand reviewers cannot provide us with further discussions. Prior to the reversion of the ratings, we had not received feedback from reviewers 9a93, AF2y and Q2KJ regarding our responses. We would like to affirm that our revisions have adequately addressed the concerns raised. We sincerely thank all the reviewers for their valuable comments, which has been instructive in helping us improve our paper.

---

### Note · Authors · 2025-12-29

I have read and agree with the venue's withdrawal policy on behalf of myself and my co-authors.